# Inhibition and transport mechanisms of the ABC transporter hMRP5

Ying Huang[1,2,7], Chenyang Xue [1,2,7], Ruiqian Bu[1,2,7], Cang Wu [1,2,7], Jiachen Li[3], Jinqiu Zhang [3], Jinyu Chen[3], Zhaoying Shi[4], Yonglong Chen [4], Yong Wang [3,5] ✉ & Zhongmin Liu [1,2,6] ✉

Human multidrug resistance protein 5 (hMRP5) effluxes anticancer and antivirus drugs, driving multidrug resistance. To uncover the mechanism of hMRP5, we determine six distinct cryo-EM structures, revealing an autoinhibitory N-terminal peptide that must dissociate to permit subsequent substrate recruitment. Guided by these molecular insights, we design an inhibitory peptide that could block substrate entry into the transport pathway. We also identify a regulatory motif, comprising a positively charged cluster and hydrophobic patches, within the first nucleotide-binding domain that modulates hMRP5 localization by engaging with membranes. By integrating our structural, biochemical, computational, and cell biological findings, we propose a model for hMRP5 conformational cycling and localization. Overall, this work provides mechanistic understanding of hMRP5 function, while informing future selective hMRP5 inhibitor development. More broadly, this study advances our understanding of the structural dynamics and inhibition of ABC transporters.

Multidrug resistance-associated proteins (MRPs) are a group of transporters that belong to the ABCC subfamily of the ATP-binding cassette (ABC) transporters. Nine MRPs have been identified in humans, encoded by *ABCC1-6*, and *ABCC10-12*[1]. MRPs play a key role in mediating chemotherapy resistance by translocating anticancer drugs out of cells, which subsequently lowers the intracellular drug concentration, thereby becoming a major obstacle to effective cancer treatment. Human MRP5 (hMRP5) was identified as an MRP based on its similarity to the extensively studied MRP1[2], a representative family member that pumps out many antineoplastic agents[3]. However, hMRP5 is structurally and functionally distinct from MRP1. Unlike MRP1, which has an additional N-terminal transmembrane domain 0 (TMD0), hMRP5 contains a flexible N-terminal loop comprising Met1 to

Thr94 (Fig. 1a) that precedes the lasso domain in the primary amino acid sequence. Accumulating evidence supports that the N-terminal region (including TMD0 or lasso domain) is involved in the protein interactions and membrane traffic machinery in the ABCC family[4,5], indicating that the N-terminal loop is essential for the function of hMRP5. Moreover, hMRP5 has a regulatory motif (R motif) in its first nucleotide-binding domain (NBD) (Fig. 1a), and the counterpart region within CFTR NBD1 also contains a similar but shorter amino acid sequence called regulatory insertion (R insertion), whose phosphorylation could serve as a regulator of the conformational switch[6]. However, the biological functions of hMRP5 R motif remain elusive. In addition, glutathione affects the ability of MRP1 but not MRP5 to translocate substrates[7]. These factors suggest hMRP5 might work

[1]Shenzhen Key Labortory of Biomolecular Assembling and Regulation, School of Life Sciences, Southern University of Science and Technology, Shenzhen, 518055 Guangdong, China. [2]Department of Immunology and Microbiology, School of Life Sciences, Southern University of Science and Technology, Shenzhen, 518055 Guangdong, China. [3]College of Life Sciences, Zhejiang University, Hangzhou 310027, China. [4]Department Of Chemical Biology, School of Life Sciences, Southern University of Science and Technology, Shenzhen, 518055 Guangdong, China. [5]The Provincial International Science and Technology Cooperation Base on Engineering Biology, International Campus of Zhejiang University, Haining 314400, China. [6]Institute for Biological Electron Microscopy, Southern University of Science and Technology, Shenzhen, 518055 Guangdong, China. [7]These authors contributed equally: Ying Huang, Chenyang Xue, Ruiqian Bu, Cang Wu. ✉e-mail: yongwang_isb@zju.edu.cn; liuzm@sustech.edu.cn

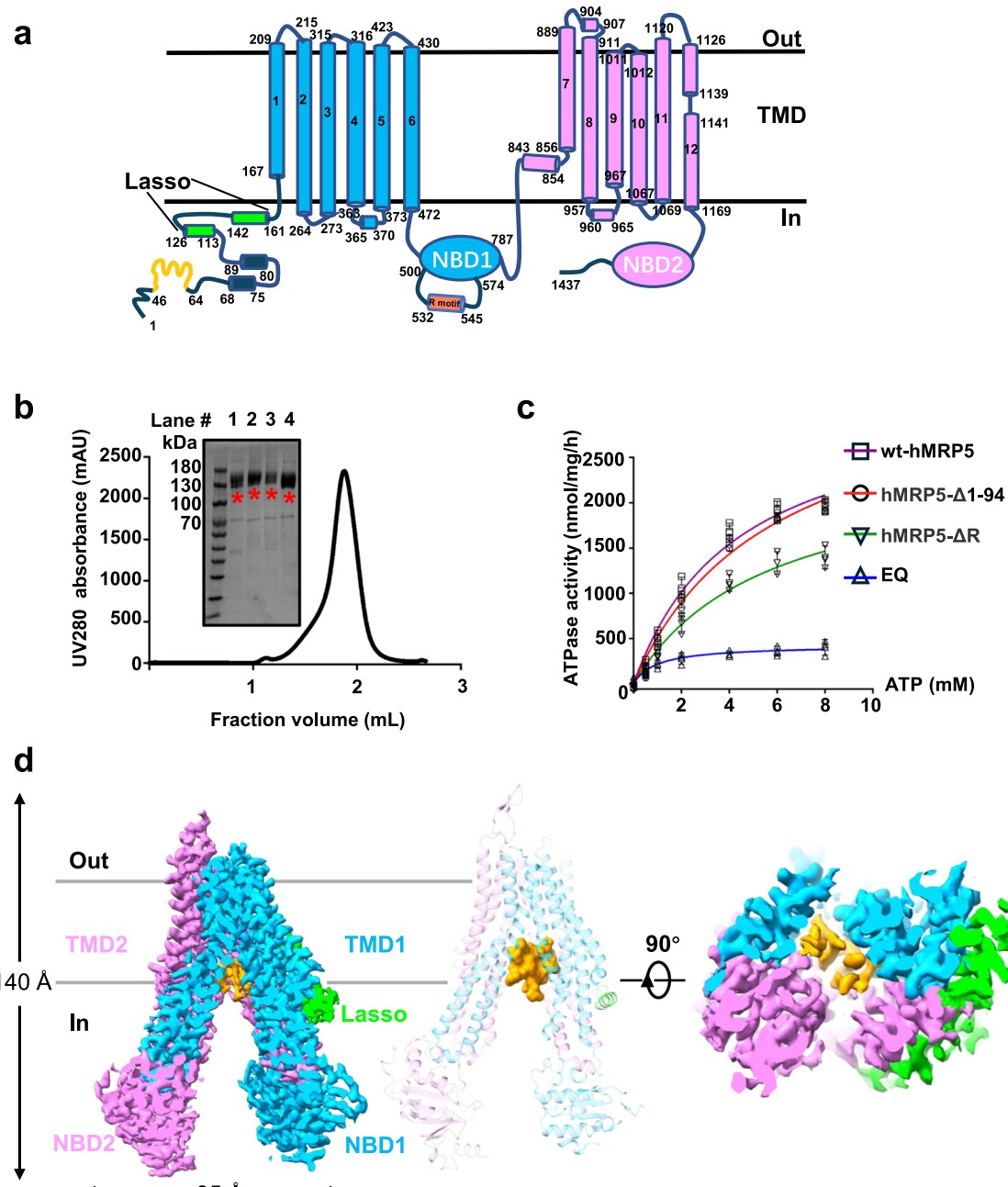

**Fig. 1 | Functional characterization and overall structure of wt-hMRP5.**
**a** Topology diagram of wt-hMRP5. Lasso domain, half 1 (transmembrane domain [TMD] 1 and nucleotide-binding domain [NBD]1), and half 2 (TMD2 and NBD2) are colored lime, deep sky blue, and violet, respectively. The special N terminal and R motif are colored orange and light coral, respectively. **b** The purified protein and size exclusion chromatography profiles of hMRP5 using a Superose 6 Increase 5/150 GL SEC column. The chromatogram is representative of >3 independent experiments with similar results. The purified protein, including hMRP5-Δ1-94, wt-hMRP5, E1355Q hMRP5 (EQ), and hMRP5-ΔR (lane 1–4), were subjected to SDS-PAGE and visualized by coomassie blue staining. The target protein is marked with an asterisk. The gel images are representative of >3 independent experiments with similar results. **c** ATPase activity of detergent-solubilized wt-hMRP5, hMRP5-Δ1-94, EQ and hMRP5-ΔR. The data points represent the means of three independent measurements and the error bars indicate the mean ± SD. Lines are fitting by nonlinear regression of the Michaelis-Menten equation. **d** Cryogenic electron microscopy map of wt-hMRP5 (left). Cartoon representation of wt-hMRP5 (middle). Bottom view of cryogenic electron microscopy map of wt-hMRP5 (right). Lasso domain is colored lime and the two halves of hMRP5 are colored deep sky blue and violet for half 1 and half 2, respectively. The N terminal density is colored orange. All structural figures were prepared using ChimeraX 1.4. Contour level is 0.246.

differently from the well-studied ABCC transporters, including MRP1 (ABCC1)[8] and MRP4 (ABCC4)[9], which begin the transport cycle with an inward-open and competent substrate binding pocket conformation.

Human MRP5 was demonstrated to play a critical function in regulating cellular cyclic GMP (cGMP) homeostasis[10], which is essential for many physiological processes, including the maintenance of vascular smooth muscle tone and cardiac contractility. hMRP5 also influences signaling transduction pathways and may contribute to drug resistance in anticancer and antivirus therapy[11–14]. From a clinical oncology perspective, hMRP5 can directly contribute to cancer drug resistance through the efflux of many anticancer drugs, such as methotrexate, cisplatin, purine analogs, pyrimidine analogs, doxorubicin, and antifolates from cells[15–18]. In addition, hMRP5 has been shown to transport monophosphate nucleotide analogs, such as

9-(2-phosphonomethoxyethyl) adenine, leading to antiviral multidrug resistance[18]. The broad spectrum of substrates that hMRP5 can transport makes it an essential transporter for the dissemination of endogenous and exogenous substrates. However, the process by which hMRP5 transports these substrates remains unclear due to the lack of solved molecular structures.

Studies have shown that long-term cisplatin treatment of lung cancer patients leads to a significantly increased expression of hMRP5[19]. Patients with high hMRP5 levels tend to have a worse survival rate than those with low expression[20–22]. Further, over-expression of hMRP5 is a strong predictor of poor outcomes in cancer patient cohorts[19,22,23]. Importantly, reducing the expression of endogenous MRP5 can notably increase the intracellular concentrations of various drugs like pemetrexed and 5-fluorouracil and its metabolites[24,25]. Meanwhile, combining MRP inhibitors with anticancer drugs has also been shown to increase treatment efficacy in cancers such as hepatocellular carcinoma, colorectal adenocarcinoma, lymphoblastic leukemia, and breast cancer[26–28]. Small molecule compounds like trequinsin, probenecid, sulfinpyrazone, MK-571, zaprinast, and sildenafil, have been reported to inhibit the transport activity of MRP5[25,29,30]. However, these existing inhibitors suffer from low affinity and efficiency, poor specificity and safety profiles, and unclear mechanisms of action. In contrast, peptide inhibitors have advantages such as high affinity and selectivity due to more extensive binding interfaces and defined conformations[31–33]. For example, a recently developed macrocyclic peptide antagonist of MRP1 called CPI1 shares mechanistic similarities with the high-affinity herpes simplex virus inhibitory protein ICP47, which inhibits the bacterial ABC transporter TAP (transporter associated with antigen processing)[34,35]. However, to date, no peptide inhibitors have been developed specifically against hMRP5.

In this work, we determine high-resolution structures of hMRP5 in various conformations using cryogenic electron microscopy (cryo-EM) to elucidate the working mechanism of hMRP5 and enable the development of specific hMRP5 inhibitors, thereby providing a clear structural understanding of hMRP5 substrate transport. Our analyses reveal that hMRP5 has a N-terminal fragment-blocked transport mechanism, distinct from other ABC transporters, and the peptide-blocked hMRP5 structure serves as an template for designing specific hMRP5 inhibitors. Our systematic structural characterization provides an understanding of hMRP5 auto-inhibition and lays the foundation for developing targeted inhibitors to overcome multidrug resistance.

## Results
### Structural features of hMRP5
To elucidate the working mechanism of hMRP5, we firstly purified 2×strep-tagged full-length hMRP5 with high homogeneity and substantial ATPase activity (Fig. 1b, c) (see more details in the method session). The following single-particle reconstruction with cryoSPARC[36] yielded a 2.9 Å hMRP5 map (Supplementary Fig. 1a–d). Local resolution estimated by LocalResmap[37] revealed variation from ~2.5 Å to ~4.5 Å (Supplementary Fig. 1a). Strong side-chain densities enabled de novo atomic modeling of most regions of hMRP5 with the exception of some flexible regions (i.e., Met1-Thr94, Asp500-Leu574, and Leu786-Ser840) due to their poor densities (Fig. 1d).

Overall, hMRP5 displayed features typical of a nucleotide-free inward-open conformation (Fig. 1d); its height and width were ~140 Å and ~95 Å, respectively. The 12 transmembrane helices (TM1-TM12) formed two pseudo-symmetrical bundles (TMD1 and TMD2), each attached to a nucleotide-binding domain (NBD) (Fig. 1d). In addition to those of the well-solved transmembrane and NBD regions, an unexpected EM density was discovered in the central cavity of the TMDs near the cytosolic leaflet. This unassigned density was peptide-sized and -shaped (Fig. 1d). This peptide-blocked inward-open conformation structurally indicates a transport mechanism of hMRP5.

## Ser501-Arg573 is not the autoinhibitory peptide but regulates hMRP5 localization
Next, to determine if the peptide captured in the cavity of hMRP5 originated from a substrate or hMRP5 itself, we subjected hMRP5 to harsh conditions, including high salt, detergents, and ATP. The peptide-derived EM density remained, suggesting it derived from an MRP5 fragment, potentially from the flexible regions Met1-Thr94, Ser501-Arg573, or Leu786-Ser840. Sequence analysis revealed that Ser501-Arg573 is not conserved among the MRP family, forming a short helix flanked by two loops (Fig. 2a, b). This suggested a potential regulatory role of Ser501-Arg573; thus, we termed the fragment of Ser501-Arg573 as the regulatory (R) motif. To determine whether the peptide-derived EM density is from the R motif, we generated hMRP5-ΔR by removing the R motif and determined its structure by the single particle reconstruction (Supplementary Fig. 2a–c). hMRP5-ΔR displayed a relatively low ATPase activity (Fig. 1c) and adopted a similar conformation to wt-hMRP5, with a peptide still trapped in the binding pocket (Supplementary Fig. 2d). These results suggested that the R motif does not affect the organization of the TMDs and NBDs, and is not the source of the peptide EM density in the hMRP5 cavity.

To figure out the potential roles of the R motif, we analyzed the amino acid sequence and found that the R motif comprises a basic residue-enriched region and two hydrophobic patches (Fig. 2a, b, Supplementary Fig. 3a). Notably, through mass spectrometry, several phosphorylation sites were identified within the R motif, such as Ser505, Ser509, Thr513, and Ser558 (Supplementary Fig. 2e). Phosphorylation could modulate the dynamics and interactions of the R motif by adjusting its charge properties. These characteristics prompted us to hypothesize that the R motif may mediate lipid binding and membrane localization[38]. To test this, we used PIP strips to assay the binding of the purified R motif (Supplementary Fig. 2f) to various lipids. We found that the R motif had a relatively weak affinity for phosphatidylinositol 4,5-bisphosphate (PI(4,5)P2) and phosphatidylinositol 3,5-bisphosphate (PI(3,5)P2), but it strongly bound to phosphatidic acid (PA), phosphatidylinositol 3-phosphate (PI(3)P), PI(4)P, and phosphatidylinositol 5-phosphate (PI(5)P) (Fig. 2c). Of note, PA, PI(4)P, and PI(3)P are enriched in the endoplasmic reticulum (ER), Golgi apparatus (GA), and endosome, respectively[39].

Following this, our investigation extended to conducting coarse-grained molecular dynamics simulations on hMRP5 in asymmetric lipid bilayers, mimicking the ER membrane and plasma membrane (PM) environments (The ER inner leaflet was composed of 20% POPC, 30% POPE, 25% POPA, 10% POPS and 15% cholesterol, whereas the PM inner leaflet had similar levels of POPC (20%) and POPE (35%) but included POPI (5%) and more cholesterol (30%)) (Fig. 2d, Supplementary Fig. 3, Supplementary Movie 1). Notably, these compositions reflect the known fact that there are more PAs in ER membranes and more PIs and cholesterols in PM. These simulations revealed the R motif, when unphosphorylated, to be notably dynamic, engaging extensively with the surfaces of both types of membranes. A discernible difference was evident in the frequency of the R motif's interactions with the ER membrane compared to the PM. More specifically, the R motif's engagement with the ER membrane involved more frequent interactions with its positively charged areas, in contrast to its engagement with the PM. Furthermore, the R motif established a greater number of contacts with the ER membrane than with the PM when bound (Supplementary Fig. 3). This indicates that the membrane association of the R motif is likely driven by a combination of electrostatic and hydrophobic interactions, with negatively charged lipids (like PA) promoting more interactions with the R motif, potentially influencing hMRP5's cellular localization through dynamic interactions with the membranes.

To investigate whether the R motif impacts hMRP5 localization, we expressed wt-hMRP5-GFP and the hMRP5-ΔR-GFP mutant in HEK293T cells and imaged them by confocal microscopy. Compared

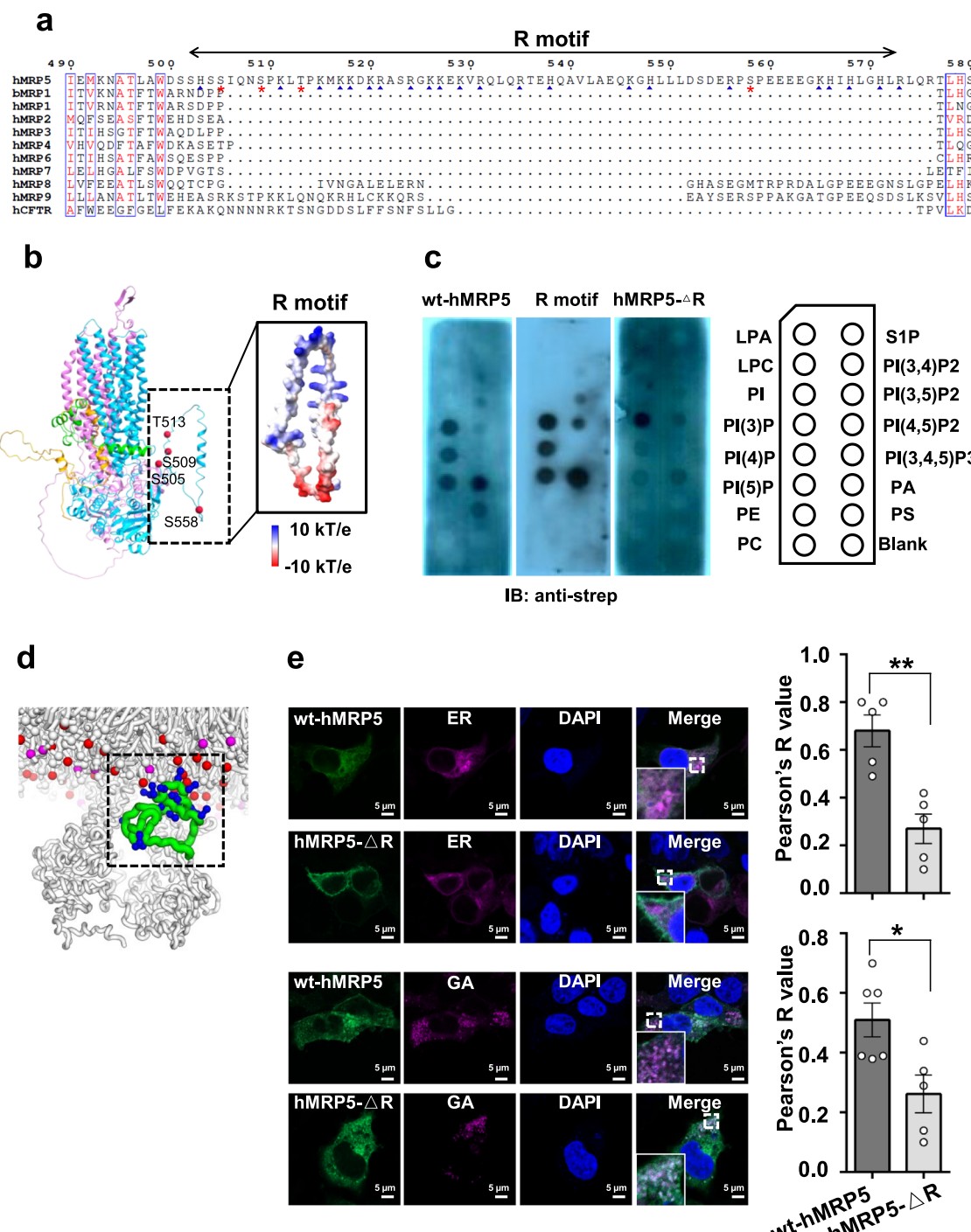

**Fig. 2 | The function of R motif. a** Sequence alignment of MRP subfamilies and human CFTR. Highly conserved residues are colored red. The phosphorylation sites are marked with red asterisks, and the positively charged residues are marked with blue triangles. **b** The predicted structure of hMRP5 from AlphaFold (https://alphafold.ebi.ac.uk/entry/O15440). Surface electrostatic potential of the R motif is shown and colored based on coulombic potential analysis. The phosphorylation sites, including S505, S509, T513, and S558, confirmed by mass spectrometry analysis are shown in red sphere. **c** Protein lipid overlay assay of recombinant wt-hMRP5-strep, R motif-strep, and hMRP5-ΔR-strep protein with membrane lipids. Recombinant proteins were incubated with a membrane lipid strip, followed by anti-strep immunoblotting. **d** Coarse-grained molecular dynamics simulation of wt-hMRP5 in an ER-like membrane. The R motif of hMRP5 is shown as a green tube,

Arginine and Lysine residues are depicted as blue sticks, and membrane lipids are represented by sticks. Additionally, lipid phosphate headgroups are highlighted as spheres, with phosphatidic acid (PA) lipid headgroups in magenta and phosphatidylserine (PS) lipid headgroups in red. **e** Fluorescent micrographs show wt-hMRP5-GFP or hMRP5-ΔR-GFP localization in endoplasmic reticulum (ER) or Golgi apparatus (GA). Quantitation of colocalization was calculated as Pearson's correlation coefficient (r) shown in right (n is representative of independent biological experiment. n = 5 for wt-hMRP5-GFP or hMRP5-ΔR-GFP localization in ER, n = 5 for wt-hMRP5-GFP localization in GA and n = 6 for hMRP5-ΔR-GFP localization in GA). Two-tailed t-test was performed. Bars indicate mean ± SEM. **P = 0.0020 and *P = 0.0172.

to wt-hMRP5, the hMRP5-ΔR mutant displayed decreased distribution to the ER and GA (Fig. 2e). This provides cellular evidence that the R motif regulates hMRP5 targeting the ER and GA. The deletion of this distinctive region alters the normal localization of hMRP5 in vesicles involved in the secretory pathway.

Collectively, these results suggest the R motif directly impacts the retention of hMRP5 in the ER and GA by mediating membrane interactions. By localizing hMRP5 to specific organelles involved in vesicular transport, the R motif likely regulates the overall transport activity and function of hMRP5.

## hMRP5 is inhibited by its own N-terminal C46-S64 fragment

Since the R motif does not contribute to the EM density in the central cavity of hMRP5, we generated a further truncation mutant hMRP5-Δ1-94 by removing the flexible region from Met1-Thr94. This deletion did not have an significantly impact on the ATPase activity (Fig. 1c) and overall inward-open conformation of hMRP5 (Fig. 3a), suggesting hMRP5-Δ1-94 retains dynamic conformational changes to enable substrate transport despite lacking the N-terminus. However, the peptide-derived density was lost in the corresponding position of hMRP5-Δ1-94 (Fig. 3a and Supplementary Fig. 4a–d). This indicates the central cavity density arose from the Met1-Thr94 flexible region, revealing an auto-blocking mechanism where hMRP5 is inhibited by its own N-terminal fragment. Of note, this is quite different from the Ycf1p (MRP1 ortholog in yeast) structure autoblocked by the regulatory domain (R domain)[40], which is topologically positioned between NBD1 and TMD2 (Supplementary Fig. 4e).

Based on the features of the auto-blocked EM density and the Met1-Thr94 peptide side chains, we were able to build an atomic model of the peptide trapped in the hMRP5 binding pocket (see more details in the method session). This peptide spanning from Cys46 to Ser64, referred to as C46-S64, was well fitted into the EM density (Fig. 3b). Spatially, C46-S64 was situated in a large (~1850 Å$^3$) binding pocket near the cytosolic leaflet (Fig. 3b), allowing extensive contact with surrounding transmembrane helices, including TM2-3, TM5-6, and TM9-12 (Fig. 3b). In the pocket, C46-S64 adopted a coiled, serpentine structure with its side chains facing the cytosol (Fig. 3b). To validate that the extra EM density originated from C46-S64, we performed all-atom MD simulations in explicit solvent (Supplementary Movie 2), which suggested that C46-S64 was stably bound in the pocket over the total simulation time of 1 μs, with very low RMSD (Fig. 3c).

The residues surrounding the C46-S64 peptide were well solved and could be mainly divided into two major groups (Fig. 3d). The first group consisted of hydrophobic amino acids, including Trp240 (TM2), Tyr244 (TM2), Phe288 (TM3), Val293 (TM3), Phe452 (TM6), Phe988 (TM9), Met992 (TM9), Phe1091 (TM11), and Phe1145 (TM12), which applied hydrophobic forces on the peptide (Fig. 3b, d and Supplementary Fig. 4f). To validate these hydrophobic interactions, we generated an hMRP5 mutant with six binding pocket mutations - Trp240A, Phe288A, Phe452A, Phe988A, Phe1091A, and Phe1145A - denoted as hMRP5-m6. High-resolution cryo-EM structure of hMRP5-m6 revealed a similar overall conformation to the wt-hMRP5 and hMRP5-Δ1-94, with separated NBDs (Supplementary Fig. 5a–e). However, the C46-S64-derived EM density was absent in the hMRP5-m6 binding pocket (Supplementary Fig. 5d), confirming the importance of hydrophobic residues in anchoring the C46-S64 peptide.

The second interaction group was composed of charged residues, including Arg286 (TM3), Glu289 (TM3), Arg344 (TM4), Lys448 (TM6), Lys455 (TM6), Arg985 (TM9), Lys1040 (TM10), Arg1096 (TM11), Arg1148 (TM12) and Glu1152 (TM12) (Fig. 3b, d, e). Notably, mass spectrometry analysis of purified wt-hMRP5 verified Ser60 is phosphorylated (P-Ser60) (Supplementary Fig. 4g), allowing P-Ser60 to contact Arg1040 and Arg1148 through charge interactions. Accordingly, Glu289, Lys1040, and Arg1096 may form salt bridges with Arg55, P-Ser60, and Glu57 of the C46-S64 peptide, respectively.

Thus, the peptide binding is reinforced by salt bridges. This electrostatic cage further stabilizes the autoinhibitory fragment within the central cavity (Fig. 3e).

Additionally, multiple hydrogen bonds further reinforced the interaction between the peptide and binding pocket (Fig. 3f). This comprehensive set of interactions contributes to the tight binding of C46-S64 within the central cavity. Sequence alignment also revealed the peptide-interacting residues are highly conserved within MRP5 across species (Supplementary Fig. 4h), indicating this autoinhibitory mechanism is shared in various organisms.

Collectively, removal of the autoinhibitory peptide does not disrupt the ATPase activity (Fig. 1c) or conformational cycling of hMRP5; thereby, the empty conformatin of hMRP5-Δ1-94 likely represents a critical apo state primed for substrate recruitment.

## A synthesized peptide inhibitor specifically blocks the hMRP5 transport pathway

We hypothesized that a synthesized peptide mimicking the C46-S64 sequence could competitively bind to the central cavity and inhibit hMRP5 transport, which could potentially reduce the efflux of chemotherapeutics by hMRP5. To test this, we designed and screened a series of N-terminal peptide fragments, including Asp5-Ser25, Cys46-Ser64, Arg40-Arg70, Asp35-Glu75, and Asp35Cys-Glu75Cys (D35C-E75C) (Fig. 4a). By screening the binding of the peptide fragments, we aimed to determine an optimal sequence for blocking hMRP5 transport.

We performed a microscale thermophoresis (MST) assay to measure the binding affinity of these peptide fragments for hMRP5-Δ1-94. Results showed that D35C-E75C had the highest affinity for hMRP5 among these peptides (Fig. 4b). This indicates that flanking residues beyond just C46-S64 also contribute to the peptide binding. Moreover, the D35C-E75C peptide displayed a higher binding affinity to hMRP5 than the well-studied substrate doxorubicin and inhibitor trequinsin (Fig. 4b). To confirm the inhibitory function of the D35C-E75C peptide, we performed the transport assay in *Xenopus laevis* oocytes, an extensively used system for studying the transport activity of ABC transporters[41]. The result showed that oocytes expressing both hMRP5 and M5PI contained a relatively higher amount of doxorubicin than oocytes expressing only hMRP5 (Supplementary Fig. 6a), supporting that the M5PI might inhibit hMRP5 transport doxorubicin.

To assess if D35C-E75C can structurally block the transport pathway, we incubated it with hMRP5-Δ1-94 at a 2:1 ratio for in vitro reconstitution. The resulting high-resolution structure of the reconstituted complex maintained an inward-open conformation with both NBDs separated (Fig. 4c and Supplementary Fig. 7a–d). Importantly, we observed clear EM density for the bound D35C-E75C peptide in the central cavity, resembling the endogenous C46-S64 fragment in wt-hMRP5. Given its ability to effectively occlude the binding pocket, we termed the D35C-E75C peptide MRP5 potential peptide inhibitor (M5PI).

Further structural analysis revealed that despite lacking phosphorylated Ser60, M5PI adopted nearly identical conformation to C46-S64 when bound in the central pocket. Additionally, beyond the C46-S64-derived density, extra EM density corresponding to Met65-Leu69 was observed (Fig. 4c), indicating these flanking residues also contact the transmembrane helices. Like the native sequence, M5PI forms an integrated network of hydrophobic, electrostatic, and hydrogen bond interactions to stably bind hMRP5 (Supplementary Fig. 7e). This mimics the tight autoinhibitory binding of the natural N-terminus fragment.

Overall, these data strongly suggest M5PI can effectively outcompete anticancer drug substrates by occupying the central cavity with high affinity. This validates the peptide inhibitor approach and demonstrates the exciting potential of blocking hMRP5 with a

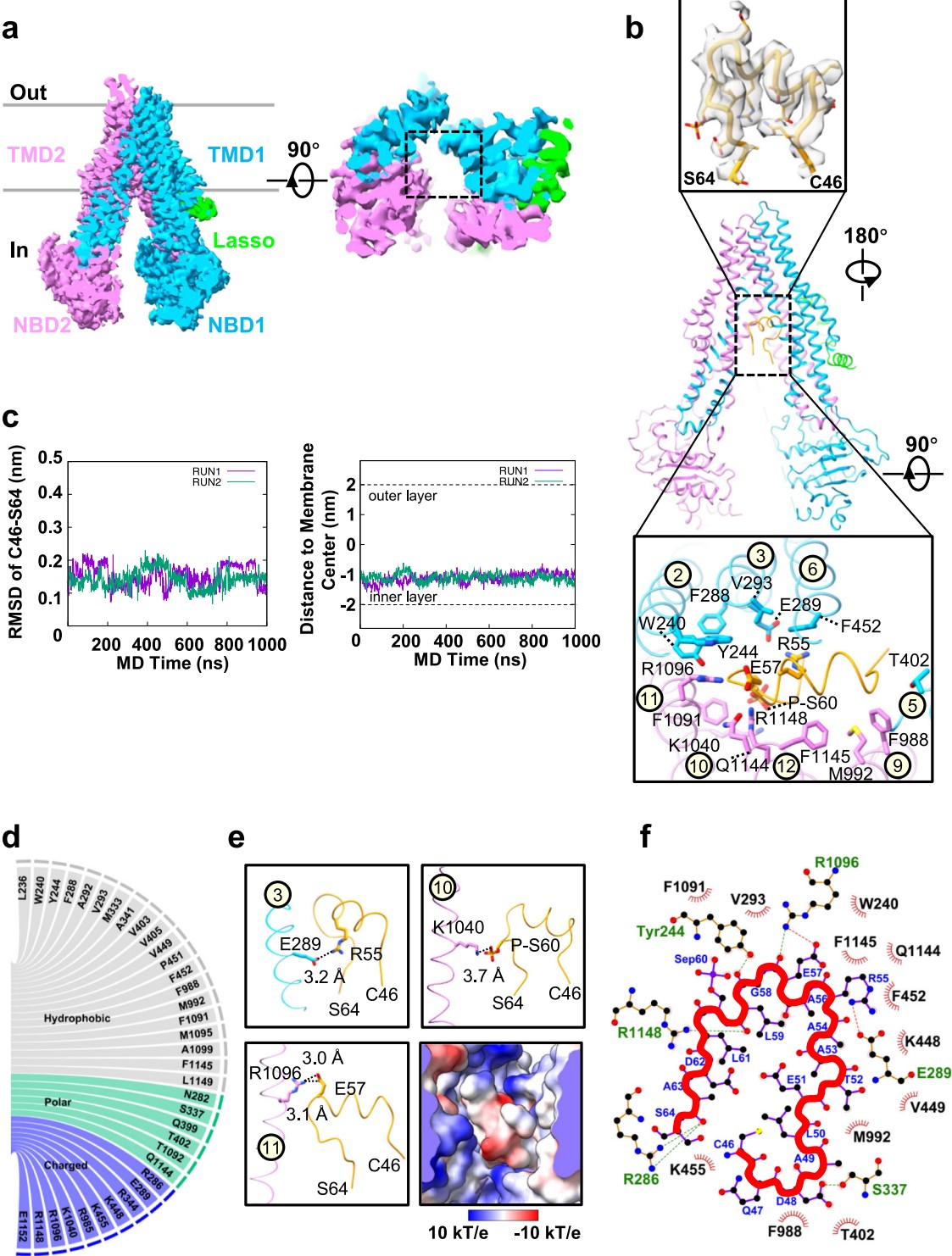

**Fig. 3 | hMRP5-Δ1-94 inward-open structure and structural basis for C46-S64 blocking hMRP5. a** Cryogenic electron microscopy map of hMRP5-Δ1-94 (left). The empty binding pocket of hMRP5-Δ1-94 is circled with a black dotted line (right). Lasso domain is colored lime and the two halves of hMRP5 are colored deep sky blue and violet, respectively. Contour level is 0.246. **b** The density of C46-S64 is shown (upper). Cartoon representation of wt-hMRP5 (middle). The binding pocket between C46-S64 and wt-hMRP5 (lower). Residues related to C46-S64 binding are shown as sticks colored by heteroatoms. Transmembrane helices (TMs) 2, 3, 5, 6, 9, 10, 11, and 12, which interact with C46-S64, are labeled. C46-S64 is colored orange. **c** Explicit-solvent all-atom molecular dynamics simulations of hMRP5 bound to the C46-S64 peptide. The left panel shows the root-mean-square deviation (RMSD) of

the C46-S64 peptide based on the backbone of the peptide itself over the course of the simulation. The right panel depicts the distance between the center of mass of the C46-S64 peptide and the center of the membrane over the simulation time. **d** Summarization of the contact sites (≤5.0 Å) between C46-S64 and wt-hMRP5. **e** The analysis of charge interaction between C46-S64 and wt-hMRP5. Surface electrostatic potential of the binding pocket of hMRP5 is shown and colored based on coulombic potential analysis. **f** Interactions between C46-S64 and wt-hMRP5. The schematic is generated by LigPlot⁺ v1.4. Each eyelash motif indicates a hydrophobic contact. Green and red dotted lines indicate hydrogen bonds and salt bridge interaction, respectively. The thick red line shows the backbone of C46-S64.

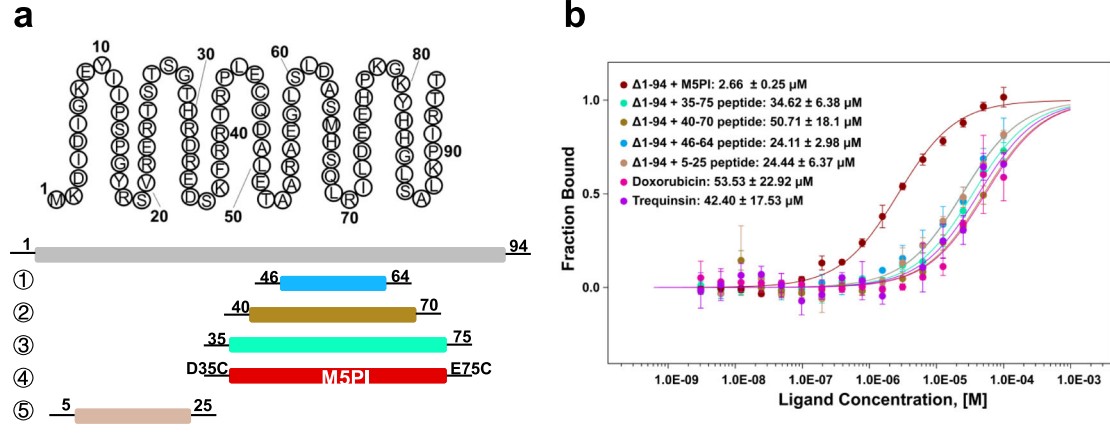

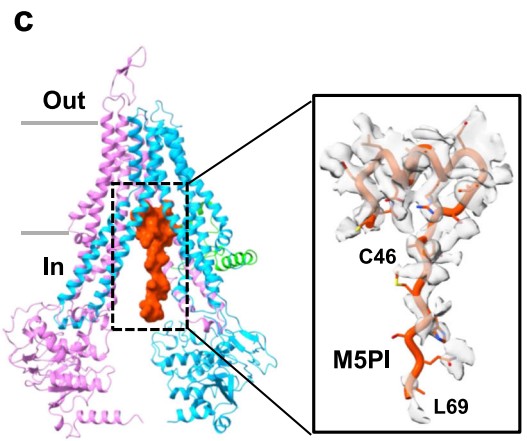

**Fig. 4 | Peptide (M5PI)-bound and ATP-bound structure. a** The amino acid sequence of N terminal (1-94) of wt-hMRP5 (upper). The domain architecture of different peptides used in the assay (lower). **b** Microscale thermophoresis (MST) measurement of the binding affinities between hMRP5-Δ1-94 and different peptides, doxorubicin and trequinsin. Data represent mean ± SEM of three independent measurements. **c** Cartoon representation of M5PI-bound structure (left). Cryogenic electron microscopy map of M5PI (orange red) (right). Contour level is

0.1. **d** Cryogenic electron microscopy map and cartoon of ATP-bound hMRP5. Two perpendicular side views are shown. The lasso domain is colored lime, and the two halves are colored deep sky blue and violet, respectively. Contour level is 0.52. ATP is represented by gray sticks colored by heteroatoms, and $Mg^{2+}$ is represented by green-yellow spheres. V shape with red dashed line shows the outward-open state of ATP-bound hMRP5. A measured opening angle is shown.

synthetic peptide derived from its own regulatory region as a novel cancer therapy.

## ATP-bound hMRP5 reveals an outward-open conformation

As an ATP-driven transporter, hMRP5 must undergo the crucial step of nucleotide binding to transport its substrates. To capture the nucleotide-bound form of hMRP5, we mutated the catalytic glutamate E1355 within the Walker B motif to a glutamine (EQ). This glutamine substitution reduces ATP hydrolysis while maintaining ATP binding capability. The E1355Q mutant was purified from the HEK293F cell line and incubated with 6 mM ATP-$Mg^{2+}$ for 1 h prior to vitrification for cryo-EM imaging. Further data processing yielded a cryo-EM map of nucleotide-bound hMRP5 at an overall resolution of 3.5 Å (Supplementary Fig. 8a–d). The structural comparison confirmed the ATP-bound hMRP5 adopted an outward-open conformation, resembling ATP-bound states of bovine MRP1 (bMRP1) instead of human SUR1 (hSUR1)[42] (Fig. 4d and Supplementary Fig. 8e).

The ATP-bound hMRP5 structure showed notable differences compared to the nucleotide-free inward-open state (Supplementary Fig. 8f). The two NBDs were bridged by two ATP molecules in a typical "head-to-tail" configuration (Supplementary Fig. 8f). Importantly, the curvature of the transmembrane helices formed an elongated cavity

across the TM region with a large volume ~4540 $Å^3$ open to the extracellular side (Fig. 4d). A thorough analysis of the ATP-bound EM map revealed that no extra EM density was observed in the transport pathway and the remaining regions of hMRP5, supporting that the auto-inhibitory C46-S64 peptide must dissociate from the binding pocket and the whole N-terminal region became very flexible. Thus, the ATP-bound structure likely represents a post-substrate extrusion state.

## Side chain orientations facilitate the transport process of hMRP5

Through a comprehensive comparative analysis of our solved hMRP5 structures, we identified key residues responsible for the rearrangement of the transport pathway during the transport cycle (Fig. 5). We found that the helix backbones remained unchanged during the transition from the peptide-autoinhibited state to the apo form (Fig. 5a). However, the side chain orientations of several residues differed, including Tyr244 (TM2), Glu289 (TM3), Phe452 (TM6), Phe988 (TM9), Lys1040 (TM10), and Arg1148 (TM12) (Fig. 5b, c). Notably, in the auto-inhibited state, the crowded binding pocket pushed the Phe452 benzene ring towards the cytosol (Phe452-Phe988 distance ~12.8 Å) (Fig. 5c). In the apo form, the Phe452 side chain

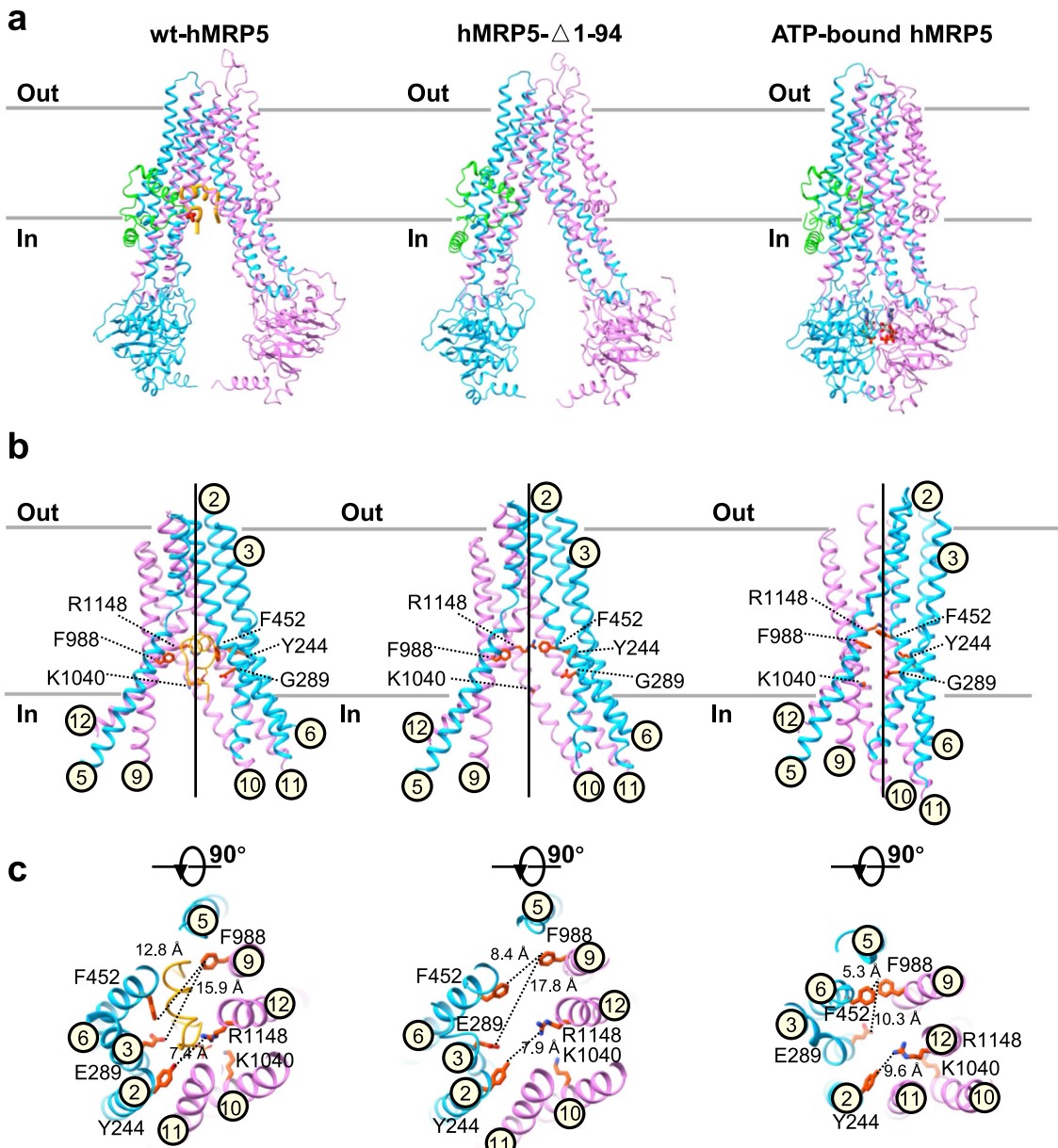

**Fig. 5 | Structural comparison among wt-hMRP5, hMRP5-Δ1-94, and ATP-bound hMRP5. a** Overall views of the wt-hMRP5, hMRP5-Δ1-94, and ATP-bound hMRP5 structures from within the plane of the membrane. A cartoon representation of wt-hMRP5 shows C46-S64 colored orange and the phosphorylation of S60 colored by heteroatoms. ATP is represented by gray sticks colored by heteroatoms, and Mg²⁺ is represented by green-yellow spheres. **b** Close-up view of the TMDs of each conformation viewed from within the plane of the membrane. Side chains of selected C46-S64 binding sites are shown as sticks and colored by heteroatoms. TM helices are numbered. **c** Extracellular view of each conformation. TM helices are shown as cartoons and numbered. Side chains of selected C46-S64 binding sites are shown as sticks colored by heteroatoms. The distances between the Phe452 of TM6 and the Phe988 of TM9, Glu289 of TM3 and the Phe988 of TM9, Tyr244 of TM2, and the Arg1148 of TM12 are indicated.

reoriented toward Phe988 (Phe452-Phe988 distance ~8.4 Å) (Fig. 5c). Furthermore, in the auto-inhibited state, Lys1040 oriented its bulky side chain vertically to the helix. In the apo form, Lys1040 rearranged towards the cytosolic side (Fig. 5b, c). In addition, upon peptide dissociation, Glu289 reorientated from the pocket center to the side (Phe988-Glu289 distance increased from 15.9 Å to 17.8 Å). Meanwhile, Arg1148 moved from the cytosolic side to the pocket center, likely due to the loss of the electrostatic interaction from phosphorylated Ser60 (Fig. 5b, c). Together, these side chain reorientations created an inward-facing empty binding pocket capable of substrate binding.

In the ATP-bound outward-open state, significant rearrangement occurred in the transmembrane regions. Specifically, TM2-3, TM5-6, and TM9-12 moved closer together during the transition from the apo

to ATP-bound state (Fig. 5b). Notably, the unique kink in the middle of TM5 disappeared as TM5 became a straight helix (Fig. 5b), reminiscent of the kinked TM4 and TM10 of ABCB1[43], which were re-arranged to a straighter conformation during the apo to ATP-bound transition. Consequently, shorter distances between pocket residues Glu289 (TM3), Phe452 (TM6), Phe988 (TM9), and Arg1148 (TM12) disrupted the substrate binding pocket, likely expelling any bound substrate from hMRP5 (Fig. 5b, c). Collectively, these conformational changes reveal how the transport pathway of hMRP5 transitions between different states.

## A working model of hMRP5
Through extensive analysis of the structural data and R motif function, we have uncovered a working mechanism for hMRP5 distinct

from all other reported ABC transporters (Fig. 6). Here, we proposed a model details how hMRP5 transports substrates across the membrane and suggests a putative peptide drug that could inhibit hMRP5 transport activity.

Firstly, newly translated hMRP5 is located on the ER surface, where the R motif can interact with the ER-dominant phospholipid PA, thereby facilitating the retention of hMRP5 on the ER surface. Next, hMRP5 moves to the GA for further maturation, aided by the interaction between the R motif and the GA-dominant PI(4)P. Finally, hMRP5 is translocated to the plasma membrane, where it can pump out endogenous and exogenous substrates.

The transport mechanism of hMRP5 involves several key conformational states. Initially, the substrate binding pocket is auto-blocked by the C46-S64 peptide (inward-open auto-blocked state), preventing substrate entry. Upon release of the blocking C46-S64 peptide, an empty pocket is formed capable of binding ligands (inward-open apo state), analogous to the first step for ABCC1[8] and ABCC7[44]. Substrates like cGMP and 6-Mercaptopurine can then be recruited to the binding pocket (substrate-bound inward-open state), in a conformation reminiscent of LTC4-bound bMRP1 (PDB:5UJA)[8]. Of note, inhibitors such as M5PI can outcompete anticancer/antiviral drugs for the binding pocket in this state. Consequently, inhibitors block hMRP5's transport pathway (M5PI-bound inward-open state), yielding an increased intracellular drug concentration and greater therapeutic efficacy. After substrate binding, ATP induces NBD dimerization, collapsing the pocket via TMD movement while creating a low-affinity outward-open pocket to release substrates (ATP-bound outward-open state). This transitions to an ATP-bound occluded state after substrate release, similar to hSUR1 (PDB: 6C3O). Finally, ATP hydrolysis reopens the closed NBDs, releasing Pi and ADP, which reopens the pocket but is auto-blocked by C46-S64 until a new ligand binds, beginning a new cycle.

## Discussion

In this study, the well-solved cryo-EM map of hMRP5 enabled unambiguous building of the atomic model of the C46-S64 peptide binding to hMRP5. This helped uncover the distinct working mechanism of hMRP5. Notably, this structure demonstrating auto-blockage by its own N-terminal fragment.

While the C46-S64 peptide is anchored in the binding pocket by numerous interactions, its density in the cryo-EM map is weaker than surrounding residues mainly because of its high heterogeneity (Supplementary Fig. 1e). Intriguingly, the Asp5-Ser25 peptide also displayed weak hMRP5 affinity, indicating that multiple segments of the N-terminal region are involved in self-interactions with hMRP5. This observation is somewhat similar to what has been seen with ABCB1[45] and ABCG2[46,47], where both proteins have demonstrated multiple specific binding sites. Additionally, a notable kink in the middle of TM5 contributes to the formation of the central cavity. This specific TM5 kink is not found in other MRP family members but in their orthologs (Supplementary Fig. 1f, g), suggesting its potential role in the translocation of hMRP5 substrates. Collectively, the unique features of the intrinsic peptide block and the TM5 kink provide key structural insights into the distinct mechanism of hMRP5.

The mechanism by which the inhibitory peptide is released from hMRP5 remains to be elucidated through further investigation. Here, several potential mechanisms for the release of C46-S64 can be proposed. One possibility is that post-translational modifications (PTMs), such as dephosphorylation, could induce conformational changes in the protein structure, weakening the interaction between the C46-S64 sequence and its binding pocket. Alternatively, the binding of allosteric effectors or partner proteins could also serve as a trigger. These molecules could either directly compete for the binding site or induce structural alterations that destabilize the C46-S64 peptide's interaction, thereby facilitating its release.

MD simulation analysis revealed that the R motif preferred interacting with negatively charged lipids such as PA, PI, and PS in both ER and PM models (Supplementary Fig. 3h, i). However, experimental results with wild-type hRMP5 indicated a lower affinity for PS compared to PA, with the isolated R motif rarely interacting with PS (Fig. 2c). The observed selectivity difference between simulation results and experimental findings may arise from limitations in both the coarse-grained modeling and the lipid strip assay for accurately assessing molecular interactions. Of note, the R motif is embedded with the NBD1 of hMRP5, distinct from the well-studied regulatory domain (R domain) of CFTR and Ycf1, which is located between the NBD1 and TMD2 and prevents NBD dimerization and channel opening[40,44,48]. In contrast, the R motif residues within the first NBD of hMRP5 and regulates its cellular distribution by interacting with phospholipids. Moreover, the further chemical modification prediction revealed that multiple residues of hMRP5 R motif may also suffer from phosphorylation and glycosylation (Supplementary Fig. 2g). The subsequent sequence and mass spectrometry analysis verified that the R motif is highly conserved and could be phosphorylated but not glycosylated with high potential, ruling out that glycosylation plays a regulatory role in the R motif. However, it remains unclear how the phosphorylation modifications affect the binding of R motif to lipids. The impact of these phosphorylations on the R motif to phospholipids, the kinases involved, and the functional consequences for hMRP5 require further investigation.

Based on sequence analysis, hMRP8 and hMRP9 each contains a motif in the counterpart region to hMRP5; however, both of those motifs are composed of fewer than 50 amino acids, which are shorter than the hMRP5 R motif and lack hydrophobic patches (Fig. 2a, and Supplementary Fig. 3a). Interestingly, the first part of the putative hMRP9 R motif is enriched in positively charged residues reminiscent of the hMRP5 motif; however, hMRP8 has fewer positively charged residues. We speculate that the R motif of hMRP9, but likely not hMRP8, may have a phospholipid-binding potential, however, with a different binding mode from hMRP5's R motif.

Although it is known that hMRP5 transports various substrates, it has long been unclear how these substrates are recognized and trapped in the binding pocket of hMRP5. We incubated hMRP5-Δ1-94 with various substrates, including progesterone, testosterone, and cGMP, in an effort to trap the substrate-bound hMRP5 structure. However, the MST assay revealed that all of these reported substrates displayed a relatively low binding affinity to hMRP5 (Supplementary Fig. 6b, c), making it challenging to obtain a stable substrate-bound hMRP5 complex. Thus, the structural basis of hMRP5 trapping of these substrates remains to be determined.

Notably, M5PI exhibited stronger hMRP5 affinity than selected anticancer drugs, suggesting it could serve as a promising template for peptide inhibitors that outcompete anticancer drugs. To justify the use of inhibitory specificity against hMRP5, we also tested the binding affinity of M5PI peptide to hMRP4; results demonstrated that hMRP4 could capture prostaglandin E1 (PGE1) but rarely M5PI, further supporting the inhibitory specificity of M5PI against hMRP5 (Supplementary Fig. 6d). Totally, our synthesized peptide displayed high hMRP5 specificity, making it a good starting point for the development of drugs targeting hMRP5.

To improve stability, the termini of the Asp35-Glu75 peptide were substituted with cysteines, aiming for peptide cyclization. While the actual cyclization and formation of an inter-molecular disulfide bridge were not confirmed yet, the resulting Cys35-Cys75 peptide exhibited a higher affinity for hMRP5 compared to the original Asp35-Glu75 peptide. The segment from Cys46 to Arg70 showed clear density within the central cavity of hMRP5, whereas the rest of the Cys35-Cys75 peptide demonstrated greater flexibility, evidenced by its weaker density.

Therapeutic peptides have inherent disadvantages, such as poor in vivo stability and weak membrane permeability[49,50]. To harness their

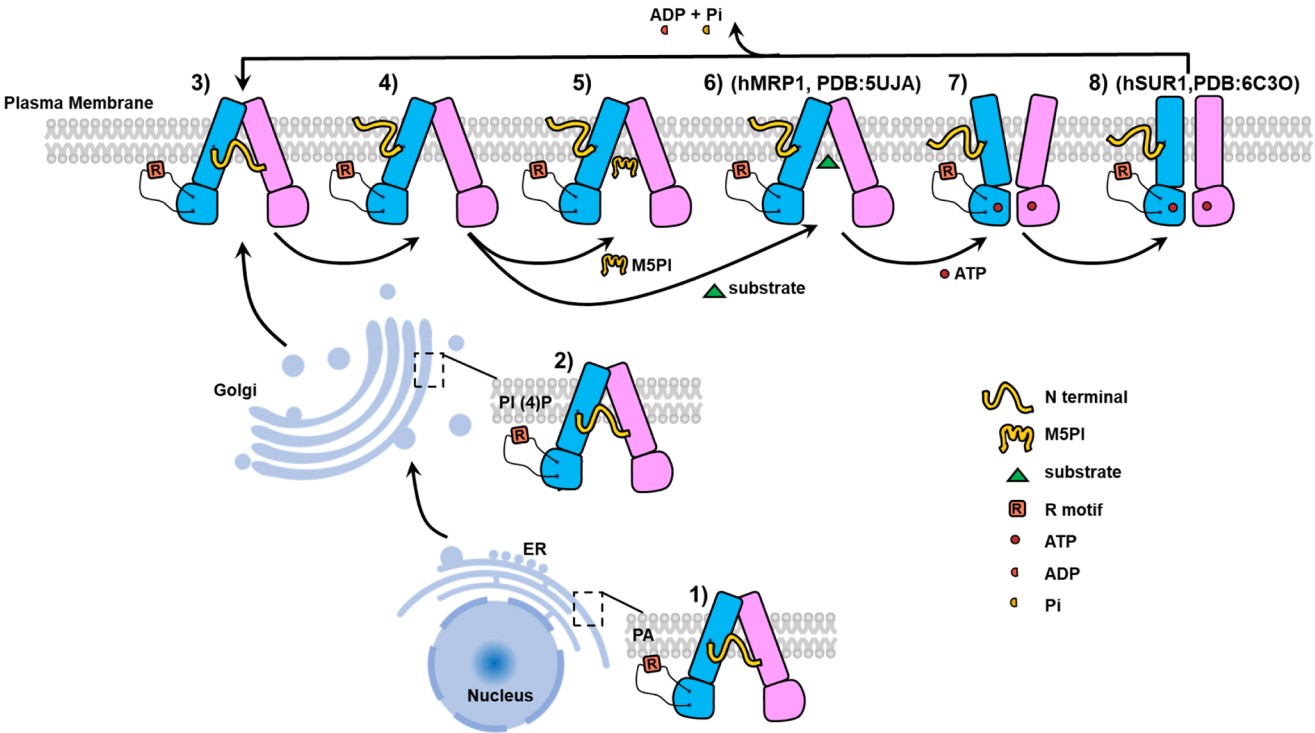

**Fig. 6 | Proposed transport mechanism of hMRP5.** Schematic of the proposed hMRP5 transport cycle. ATP is shown as a red circle. ADP and Pi are shown as cyan and yellow semicircles, respectively. Major conformational states are represented by numbers. 1) Synthesis of hMRP5 in endoplasmic reticulum. 2) Processing and packaging of hMRP5 in Golgi apparatus. 3) wt-hMRP5 showing the inward-occluded conformation with C46-S64 binding in the binding pocket. 4) Apo form of hMRP5 showing the inward-open conformation and the C46-S64 swing away from the binding pocket. 5) M5PI-bound hMRP5. 6) Substrate-bound hMRP5 is referred to MRP1-LTC4 (PDB:5UJA). 7) ATP-bound hMRP5 showing the outward-open conformation. 8) Outward-occluded conformation upon ATP-bound is referred to the ATP-bound hSUR1 (PDB: 6C3O).

potential for cancer therapy, it is crucial to tackle issues related to specificity and membrane-crossing capabilities. We expect that the application of AI technologies, such as RFdiffusion[51], for the de-novo design of peptides could significantly enhance their pharmacological effects, with their effectiveness subsequently verified via cellular tests. In summary, the auto-blocked structural basis and the herein found biological function of the R motif highly advanced our understanding of the transport mechanism of ABC transporters; moreover, our data provided us a clue for developing peptide drugs specifically targeting hMRP5, shedding light on the treatment of hMRP5 mediated drug resistance.

## Methods

### Study approval
All animal procedures were conducted following international standards and were approved by the Animal Care and Use Committee of the Southern University of Science and Technology (Approval No: SUSTech-JY202307040). *Xenopus laevis* (two to three years old) females were used for oocyte isolation.

### Cell culture
HEK293F cells and 293T cells were gifts from Prof. Hong-Wei Wang (Tsinghua University, Beijing, China). HEK293F cells were cultured in SMM 293-TII medium (Sino Biological) in the following conditions: 37 °C and 5% $CO_2$ in a shaker. 293T cells were grown in DMEM (Hyclone) containing 10% fetal bovine serum and 1% penicillin-streptomycin in the following conditions: 37 °C, 80% humidity, and 5% $CO_2$.

### Mutagenesis
Mutations, including EQ (E1355Q/D716C/H1386C) and hMRP5-m6 (W240A/F288A/F452A/F988A/F1091A/F1145A), were created with the Fast Mutagenesis System kit (Transgen) using standard two-step PCR and verified by DNA sequencing (Shenggong Biotech). Information about the plasmids and primers is provided as Supplementary Data 1 and 2.

### Protein expression and purification
The codon-optimized full-length human *ABCC5* gene encoding the MRP5 protein (UniProt ID: O15440) was cloned into the pCAG vector with C-terminal 2×strep tags via homologous recombination with a ClonExpress Ultra One Step Cloning Kit (Vazyme Biotech). The other vectors used in this study were constructed with the same protocol.

When the cell density reached ~2.0 × 10⁶ cells/ml, ~1.5 mg plasmids were transfected into 800 ml HEK293F cells with 4.5 mg linear polyethylenimine (Polysciences). The transfected cells were grown at 37 °C for 48 h. After centrifugation at 1500 × *g* for 5 min, the cell pellets were resuspended and washed with 1×phosphate-buffered saline (PBS), quickly frozen in liquid nitrogen, and stored at −80 °C until further use.

For purification, the following steps were performed at 4 °C. Cell pellets were thawed and gently resuspended in lysis buffer containing 100 mM Tris-HCl (pH 8.0), 150 mM NaCl, 1 mM phenylmethylsulfonyl fluoride, 2 mM dithiothreitol (DTT), 1% (w/v) DDM (Anatrace), and 0.2% (w/v) CHS (Sigma) with an EDTA-free protease inhibitor cocktail (Roche). Two hours later, the cell lysates were subjected to centrifugation at 50,000 × *g* for 1 h and the supernatants were transferred to strep-Tactin (IBA Lifesciences), and further rotated at 4 °C for 1.5 h. Then, the resin was collected and rinsed with 20 column-volumes of wash buffer containing 25 mM Tris-HCl (pH 8.0), 150 mM NaCl, 10% (v/v) glycerol, 2 mM DTT, and 0.06% (w/v) digitonin (Apollo Scientific). The protein was eluted with an elution buffer containing 25 mM Tris-HCl (pH 8.0), 150 mM NaCl, 5% (v/v) glycerol, 2 mM DTT, and 0.06% (w/v) digitonin supplemented with 50 mM biotin. The protein eluent

was concentrated using a 100-kDa cut-off Centricon filter (Millipore) and further purified by SEC using a Superose 6 Increase column (GE Healthcare) in an SEC buffer containing 25 mM Tris-HCl (pH 8.0), 150 mM NaCl, 2 mM DTT, and 0.06% (w/v) digitonin. The peak fractions were pooled and concentrated for further biochemical studies or cryo-EM experiments.

The wildtype and mutant proteins used in this project were expressed and purified using the protocol described above. For the E1355Q mutant, 2 mM ATP and 2 mM MgCl$_2$ were added in all buffers during purification. For the R motif fragment, 1% Triton instead of DDM was used in the lysis buffer, and digitonin was removed in the following wash and elution buffer. The original protein gels are provided in Source Data 1, Source Data 2 or Supplementary Information file.

## ATPase activity assay
The ATPase activities of wt-hMRP5 and its mutants, including E1355Q, hMRP5-Δ1-94 and hMRP5-ΔR, were assessed with a commercially available kit that measures the inorganic phosphate (Pi) released from ATP, according to the kit protocol (Nanjing Jiancheng Bioengineering Institute, China) in 96-well plates. To measure the ATPase activity of hMRP5 in varying ATP concentrations, we added the proteins (final concentration, ~1 μM) to the reaction buffer containing 25 mM Tris-HCl (pH 8.0), 150 mM NaCl, 2 mM DTT, 0.06% (w/v) digitonin (Apollo Scientific) and 2 mM MgCl$_2$. ATP was supplemented at a series of concentrations to assay ATPase activity. The reactions were mixed with matrix buffer, and incubated at 37 °C. The supernatants were collected after centrifugation at 1200 × $g$ for 10 min, and then further incubated with chromogenic agent and termination solution, sequentially. Finally, the amount of released Pi was detected by measuring optical density at 636 nm and quantified through calculation. Statistical analysis of the ATP activity assay results were fit by nonlinear regression to the Michaelis-Menten using GraphPad Prism. Source data of the ATPase activity is provided as Source Data 3.

## Lipid strip assay
Membrane lipid binding analysis of strep-tagged wt-hMRP5, hMRP5-ΔR, and the R motif was conducted using Membrane Lipid Strips (Echelon Bioscience), with each spot containing 100 pmol of the indicated lipid. The membranes were blocked in PBS containing Tween 20 (PBST) and 3% bovine serum albumin for 1 h at room temperature, then incubated with 0.25 μg/ml target protein for 1 h in blocking buffer. After washing three times with PBST, the membranes were blotted with anti-strep antibody (Shenggong Biotech) at 1:5000 dilution.

## Transfection and immunostaining assays
293T cells were grown in confocal dishes and transfected with plasmid (~1200 ng) in opti-MEM (Invitrogen Life Technologies) using Lipofectamine 2000 (Invitrogen Life Technologies) when the cell density reached 50–60%. The plasmids, including wt-hMRP5-GFP, hMRP5-ΔR-GFP, ER-mCherry, and GA-mCherry, were used in immunostaining assays. After transfection for 24 h, the medium was removed, and the cells were fixed in 4% (w/v) paraformaldehyde for 10 min at room temperature. After washing three times with PBS, the samples were mounted in fluorescent mounting medium with 4,6-diamidino-2-phenylindole (DAPI, ZSGB BIO) and imaged with a Zeiss Imager at excitation wavelengths of 488 nm for GFP, 568 nm for mCherry, and 405 nm for DAPI. We performed colocalization quantification using Pearson's correlation coefficient with ImageJ software. Statistical analysis were performed using GraphPad Prism. Source data of the quantification is provided as Source Data 4.

## MST binding assay
We monitored the binding of different peptides to digitonin-purified hMRP5-Δ1-94-8xHis by MST assay (Monolith NT). HEPES buffer was used to replace Tris buffer during protein purification in the MST assay. The purified protein was mixed with RED-tris-NTA 2$^{nd}$ Generation dye (NanoTemper, MO-L018). Next, serially diluted peptides were mixed with the labeled protein. The mixtures were incubated for 15 min at room temperature, then the capillaries were loaded and the samples were assessed at 3% LED/excitation power and medium MST power. All peptides used in this study were synthesized in GL Biochem (Shanghai). The peptide sequence is provided as Source Data 5. The MST data were analyzed with MO.Affinity Analysis software.

## Transport assay in *Xenopus laevis* oocytes
The transport assay was essentially done and modified as described[41]. In brief, the cRNAs of C-terminal GFP-tagged hMRP5 and C-terminal mcherry-tagged M5PI were prepared from the linearized pcs2CAG vector DNA templates using the mMESSAGE mMACHINE T7 transcription system (Thermo Scientific, AM1344). *Xenopus* oocytes were injected with approximately 30 nl of cRNA (GFP-tagged hMRP5 and mcherry-tagged M5PI have the equal quantity 30 ng) or water after maintained in ND96 solution (96 mM NaCl, 2 mM KCl, 1 mM MgCl$_2$, 1.8 mM CaCl$_2$, 5 mM HEPES, pH 7.4) overnight. After 3 days of culture, the GFP and mcherry fluorescence of oocytes was detected using Nikon Confocal A1R, and oocytes showing similar fluorescence intensity were selected for assay. Then, oocytes were pre-incubated in Kulori buffer (90 mM NaCl, 1 mM KCl, 1 mM MgCl$_2$, 5 mM HEPES, pH 7.4) for 5 min before being transferred to 1 ml Kulori buffer containing 200 μM doxorubicin for 180 min. The assay was stopped by removing the buffer and then washed with Kulori buffer at least 3 times and transferred into 1.5 ml tubes (3 oocytes per tube). Excess washing buffer was removed from the tube, and then 50 μl 50% methanol was added to the tube for oocyte homogenization by pipetting. The homogenized samples were centrifuged at 10,000 × $g$ for 10 min at 4 °C, and the supernatants were transferred into new tubes and incubated at −20 °C until further analysis. The samples were acquired with a Q Exactive mass spectrometer coupled with UltiMate 3000 liquid chromatography system (Thermo scientific, CA, USA) after centrifugation at 20,000 × $g$ for 15 min at 4 °C. A Hypersil GOLDTM C18 column (2.1 ×100 mm, 1.9 μm, Thermo Scientific) was used at a flow rate of 0.3 ml min$^{-1}$ for separation. Mobile phase A was 0.1% formic acid, while mobile phase B was acetonitrile modified with 0.1% formic acid. The mass spectrometer was operated in positive mode with a spray voltage of 3.5 kV. Full scan mass spectrometry data were recorded from the m/z range of 80–800. A standard dilution curve for quantification of doxorubicin was done. Statistical analysis of the transport assay results was performed using GraphPad Prism. Source data is provided as Source Data 6. The mass spectrometry data is provided as Supplementary Data 3.

## Molecular dynamics simulations
To investigate the dynamics of the R-motif in hMRP5, we first constructed an all-atom model of truncated hMRP5 construct (Δ1-94) in the uninhibited apo state using the cryoEM structure as a template. The missing intracellular and extracellular loop regions absent in the cryoEM structure were modeled as unstructured loops. This all-atom model was subsequently coarse-grained at a resolution of approximately four heavy atoms per bead using the martinize.py script. The coarse-grained hMRP5 model was then embedded into an asymmetrical lipid bilayer mimicking the ER membrane or PM. This bilayer was composed of two different leaflet compositions. In the outer leaflet, it contained a mixture of 70% 1-palmitoyl-2-oleoyl-sn-glycero-3-phosphocholine (POPC) and 30% cholesterol. For the inner leaflet of the ER-like membrane, the composition was 20% POPC, 30% 1-palmitoyl-2-oleoyl-sn-glycero-3-phosphoethanolamine (POPE), 25% 1-palmitoyl-2-oleoyl-sn-glycero-3-phosphate (POPA), 10% 1-palmitoyl-2-oleoyl-sn-glycero-3-phospho-L-serine (POPS), and 15% cholesterol. In the inner leaflet of the PM-like membrane, the mixture was 20% POPC, 35% POPE, 5% phosphatidylinositol (POPI), 10% POPS, and 30%

cholesterol. The box has dimensions of $13 \times 13 \times 16.5\,nm^3$. The membrane embedding was performed using the INSANE protocol[52] to match the hydrophobic thickness and transmembrane regions of hMRP5. The Martini3 force field parameters[53] were employed, along with elastic network restraints applied to the backbone beads of specific residues – specifically from residue 94–503, 572–790, and 840–1437 – with a force constant of $1000\,kJ/mol/nm^2$ and cutoff distances between 0.5–0.9 nm to maintain the protein secondary and tertiary structure. The R motif was specifically modeled in its unphosphorylated state, because of the absence of parameters for phosphorylated residues in the coarse-grained Martini3 model available at the time of our research. The system was solvated with standard Martini water beads, and $Na^+$ and $Cl^-$ ions were added at a concentration of 0.15 M to neutralize the charge and mimic physiological salt concentrations. The backbones of experimentally determined high-density regions in the cryo-EM map were position-restrained with a force constant of $1000\,kJ/mol/nm^2$ to maintain the native conformation. Energy minimization and standard CHARMM-GUI membrane protein equilibration steps were performed before the production run using an integration time step of 20 fs. The production runs were conducted in duplicate, with each run lasting 100 µs for the EM-like membrane model and 50 µs for the PM-like membrane model.

For the all-atom (AA) MD simulations, the hMRP5 system in the C46-S64 peptide blocked state was built based on the corresponding cryo-EM structure using the CHARMM-GUI webserver[54]. The hMRP5 structure was inserted into a $12.5 \times 12.5 \times 16.5\,nm^3$ box containing a pure POPC lipid bilayer solvated with TIP3P water molecules. The system underwent an energy minimization step using 5000 steps of the steepest descent algorithm, followed by a six-step equilibration protocol during which position restraints on the protein heavy atoms were gradually reduced. Production MD simulations were then performed under the isothermal-isobaric (NPT) ensemble at 310 K and 1 bar using GROMACS 2021.5 accelerated on GPUs[55]. For the peptide-bound hMRP5 system, we conducted two independent 1 µs production runs.

The trajectories were analyzed using GROMACS tools, getcontacts, PLUMED[56], and custom R scripts. The molecular dynamics simulations checklist is provided in Supplementary Table 1.

## Liquid chromatography–tandem mass spectrometry analysis of phosphorylated hMRP5

The protein solution samples (10 µg) were transferred into ultrafiltration tubes (Millipore) and centrifuged at $12,000 \times g$ for 20 min, then washed twice with 200 µl buffer A (8 M urea/100 mM Tris-HCl, pH 8.5). Disulfide bonds were reduced by incubation with 20 µl DTT (100 mM) solution at 37 °C for 2 h, then the samples were alkylated with 50 mM iodoacetamide in 200 µl buffer A for 30 min in the dark. Next, the samples were washed three times with 100 µl buffer A and finally washed three times with 100 µl ammonium bicarbonate solution (25 mM). The proteins were digested at 37 °C for 14 h with 100 µl trypsin (Promega) at a concentration of 0.01 µg/µl in ammonium bicarbonate solution (25 mM). After digestion, peptides were collected and dried by vacuum centrifugation in preparation for the following experiment.

The digested samples were redissolved in 0.1% formic acid/water (Nano-LC mobile phase A) and loaded for liquid chromatography–mass spectrometry analysis. The dissolved samples (2 µl) were loaded onto nanoViper C18 pre-columns (3 µm, 100 Å) and then rinsed for desalting (20 µl). The liquid phase was processed with the nanoflow HPLC Easy-nLC 1200 system (Thermo Fisher Scientific). The samples were desalted and retained on the pre-column, then separated with a C18 reverse phase column (Acclaim PepMap RSLC, 75 µm × 25 cm, 2 µm, 100 Å). The mobile phase B gradient (80% acetonitrile, 0.1% formic acid) was increased from 5% to 38% over 30 min. Mass spectrometry was performed with the

Thermo Fisher Q Exactive system (Thermo Fisher Scientific). The spray voltage was set at 2500 V in positive ion mode and the ion transfer tube temperature was set at 275 °C. Data-dependent analysis was performed using Xcalibur software[57] for profile spectrum data. The primary mass spectrometry scanning resolution was set at 70,000, and the scanning range was 350–2000 mass/charge (m/z), with an automatic gain control target of 1e6 with a maximum injection time (IT) of 100 msec. Secondary mass spectrometry was performed with an IT of 50 msec. Dynamic exclusion was used to select an m/z exclusion list for 30 s.

The mass spectrometry and tandem mass spectrometry data was searched against the Uniprot human protein database (https://www.uniprot.org/proteomes/UP000005640) with Thermo Proteome Discoverer v2.4.0.305 (Thermo Fisher Scientific). The original raw atlas files collected by mass spectrometry were processed and analyzed using PEAKS Studio 8.5 (Bioinformatics Solutions) software. The search parameters were set as follows: trypsin enzymolysis, primary mass spectrometry mass tolerance of 10 ppm, and secondary mass spectrometry in steps of 0.05 Da. Liquid chromatography-tandem mass spectrometry analysis was performed at Shanghai Shenggong Biotech. The mass spectrometry data is provided as Supplementary Data 4.

## Cryo-EM sample preparation and data collection

Freshly purified MRP5 protein was concentrated to ~5 mg/ml. Then, 5 µl proteins were added to the freshly glow-discharged (0.39 mBar air, 15 mA, 50 s) holey carbon grid (Quantifoil Au R1.2/1.3, 300 mesh). The grids were plunge-frozen in liquid ethane using an FEI Vitrobot Mark IV (Thermo Fisher Scientific) under the following conditions: waiting time, 11 s; blot time, 4 s; blot force, −2; temperature, 10 °C; and humidity, 100%. Next, the grids were transferred to and stored in liquid nitrogen until data acquisition. For the peptide M5PI-bound complex, the peptide and hMRP5-Δ1-94 were incubated together on ice at a ratio of 2:1 for 1 h before vitrification. For the EQ mutant, 6 mM ATP (Sigma) and $MgCl_2$ were added to and incubated with the protein at room temperature for 1 h before vitrification.

The prepared grids were transferred to a Titan Krios G3i microscope (Thermo Fisher Scientific), running at 300 kV and equipped with a Gatan Quantum-LS Energy Filter (GIF, slit width of 20 eV) and a Gatan K3 Summit direct electron detector in the electron super-resolution mode or a Falcon 4 electron direct detector in the electron super-resolution mode. For wt-hMRP5, imaging was performed at a nominal magnification of 81,000×, with a super-resolution pixel size of 1.072 Å, with the K3 detector. For hMRP5-ΔR, movies were recorded at a nominal magnification of 105,000×, with a super-resolution pixel size of 0.855 Å, with the K3 detector. For hMRP5-Δ1-94, imaging was performed at a nominal magnification of 105,000×, with a super-resolution pixel size of 1.29 Å, with the K3 detector. For hMRP5-m6, dose-fractionated images were recorded at a nominal magnification of 81,000×, with a super-resolution pixel size of 1.08 Å, with the K3 detector. For M5PI-bound hMRP5, dose-fractionated images were recorded at a nominal magnification of 96,000×, with a super-resolution pixel size of 0.82 Å, with the Falcon 4 detector. For ATP-bound hMRP5, image stacks were collected at a nominal magnification of 96,000×, with a super-resolution pixel size of 0.86 Å, with the Falcon 4 detector. All movie stacks were collected with EPU software with a defocus range of −0.8 to −1.2 µm. All datasets were collected using image-beam-shift multiple recording. The total accumulated dose for each specimen was ~50 $e^-/Å^2$. Each micrograph stack contains 32 frames. All EM image-related parameters for each sample are listed in Supplementary Table 2.

## Cryo-EM data processing

The single-particle analysis procedures for all states are summarized in the supplementary information (Supplementary Figs. 1, 2, 4, 5,

7 and 8). For all datasets, the dose-fractionated movies were subjected to beam-induced motion correction, using MotionCor2[58] software or patch motion correction (cryoSPARC)[36], and the contrast transfer function (CTF) parameters were estimated using patch CTF estimation (cryoSPARC). Micrographs were inspected to remove those that were not suitable for image analysis. All subsequent image processing was performed using cryoSPARC. For all datasets, 100 micrographs were selected to train a Topaz model using the "manually curate exposures" job in cryoSPARC. Particles were picked with a blob picker and extracted with a box of size of 300 pixels, Fourier-cropped to 100 pixels, and subjected to rounds of two-dimensional (2D) classification. Selected particles from the 2D classification and the 100 micrographs were used to train the Topaz model using "Topaz train" in cryoSPARC. Then, particles from all micrographs were picked with "Topaz extract" using the trained Topaz model in cryoSPARC and extracted with a box of 300 pixels. All particles were used for 2D classification. After 2D classification, particles were selected for ab-initio reconstruction to generate the initial maps in cryoSPARC. Subsequently, the particles were classified by multiple rounds of heterogeneous refinement using the initial models. The final particles selected from the heterogeneous refinement were subjected to non-uniform refinement and local and global CTF refinements to yield a final map. The overall resolution of the final map was determined by the 0.143 criterion of the gold-standard Fourier shell correlation[59]. Local resolution estimation was used to determine the local resolution of each map in cryoSPARC.

### Model building

The sequence of MRP5 was obtained from Uniprot for *Homo sapiens* (Uniprot code: O15440). The predicted structure obtained from the AlphaFold Protein Structure Database (https://alphafold.ebi.ac.uk/entry/O15440) was used for the atomic model building of wt-hMRP5, hMRP5-Δ1-94, hMRP5-ΔR, hMRP5-m6, and M5PI-bound hMRP5. For the ATP-bound hMRP5 states, the Phyre2 online server[60] was used to obtain the initial atomic model of hMRP5. For building the atomic model of the auto-inhibitory peptide, we firstly analyzed the wt-hMRP5 EM map at about 3 Å and found the auto-inhibitory EM density was derived from -18 residues and featured by obvious protruding densities of every one or two residues; moreover, the positively charged binding pocket provided a clue that a cluster of negatively charged residues must contribute to the auto-inhibitory EM density. All these helped us to deduce the auto-inhibitory EM density might be derived from the fragment of R40-R70. After repeatedly trying, we finally determined and built the initial atomic structure of the auto-inhibitory EM density. All initial atomic models were fitted to the corresponding maps; then the coordinates relative to the cryo-EM maps were saved in UCSF-Chimera[61]. Models for all structures were built after rigid-body fitting of individual domains into the corresponding maps using the initial model and rounds of coordinate adjustment and local refinement in coot[62]. Model refinement was performed using Phenix[63] (phenix.real_space_refine) with secondary structure restraints and with the refinement resolution limit set to the overall resolution of the map. Structural figures were prepared using ChimeraX.

### Reporting summary

Further information on research design is available in the Nature Portfolio Reporting Summary linked to this article.

### Data availability

Cryo-EM maps were deposited in the Electron Microscopy Data Bank under accession codes EMD-37554 (wt-hMRP5), EMD-37556 (hMRP5-Δ1-94), EMD-37555 (hMRP5-ΔR), EMD-37557 (hMRP5-m6), EMD-37558 (M5PI-bound hMRP5), and EMD-37105 (ATP-bound hMRP5). Atomic coordinates have been deposited into the PDB under accession numbers 8WI0 (wt-hMRP5), 8WI3 (hMRP5-Δ1-94), 8WI2 (hMRP5-ΔR), 8WI4 (hMRP5-m6), 8WI5, (M5PI-bound hMRP5), and 8KCI (ATP-bound hMRP5). Source data are provided with this paper. Molecular dynamics files have been uploaded to GitHub and are accessible via the following link: https://github.com/yongwangCPH/papers/tree/main/2024/MRP5. Source data are provided with this paper.

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

## Acknowledgements

We thank all of the staff members of the cryo-EM Centre at Southern University of Science and Technology, and Chunlong Guo, Zhenqian Guo, Fanhao Meng, Li Li, and the rest of the staff members at Shuimu BioSciences for their assistance with data collection. Transport assay data were obtained using equipment maintained by Southern University of Science and Technology Core Research Facilities. Yong Wang acknowledges the financial support of the National Key Research and Development Program of China (No. 2021YFF1200404), the National Natural Science Foundation of China (No. 32371300), the Zhejiang Provincial National Science Foundation of China (No. LZ24C050003) and computational support of the Information Technology Center and State Key Lab of CAD&CG at Zhejiang University. Zhongmin Liu thanks the National Science Foundation of China (No. 32000850 and No. 32371275), Shenzhen Municipal Basic Research projects (No. JCYJ20210324105007020), Guangdong Innovative and Entrepreneurial Research Team Program (No. 2021ZT09Y104), the Pearl River Talent Recruitment Program (No. 2021QN02Y449), Shenzhen Science and Technology Program (No. ZDSYS20220402111000001), Guangdong Basic and Applied Basic Research Foundation (No. 2024A1515011538) and Southern University of Science and Technology for financial support.

## Author contributions

Z.L. and Y.W. initiated and supervised the project. Y.H. and R.B. prepared and purified the proteins. Y.H, R.B., and C.X. collected the cryo-EM data. R.B performed the lipid strip assay. C.X. and C.W analyzed and performed the model building and refinement. Y.H analyzed the structures, prepared the figures and performed the experiments. Y.W., J.L., J.Z. and J.C. performed the molecular dynamics simulations and analyzed the simulation data. Y.C and Z.S prepared the materials for transport assay. Z.L. and Y.W. drafted the manuscript with help from all authors.

## Competing interests

The authors declare no competing interests.
