## [Peer Review File · Nature Communications]

Inhibition and transport mechanisms of the ABC transporter hMRP5REVIEWER COMMENTS

Reviewer #1 (Remarks to the Author):

Key Results:

The authors used SPA cryo-EM to determine structures of the human ABC transporter ABCC5. In combination with MD-simulations and structure-guided peptide-design, the authors proposed an auto-inhibitory mechanism, which intrinsically restricts transport. Furthermore, they propose, that ABCC5 function might be regulated through interactions between PI(4)P and a sequence-motif of the nucleotide binding domain.

The structures presented in the manuscript are of good to very good quality and so is the research design and the amount of data presented. In particular the authors perform many experiments to validate and identify the interaction of the transporter with the peptide (both as wt and the peptide they developed).

However, there are some major concerns that must be addressed before publication is possible, as outlined in detail below. Most pressing is that the authors do not convincingly show that the peptide really inhibits transport. Secondly, the authors utilize different detergent mixtures for the experiments. It is not unlikely that this biases the data making a comparison difficult.

The manuscript, in its current stage is extremely lengthy and major re-writing is required to make it more concise and to enable the reader to fully grasp its content. The quality of the figures is also not suitable for this journal. The authors need to remove the white space and adhere to their color code throughout. Figure 6 in particular, needs to be redone entirely. Even though the authors use cartoon representation for IF and OF conformations it is not immediately clear which conformation is meant in each specific part. Furthermore, the circular arrangement is not helping to increase clarity.

Also, I suggest that the authors refrain from using terminology like “unique” which appears several times in the manuscript. It may be novel or unprecedented but who knows if it is really unique.

As apparent from the many specific comments below the manuscript requires a thorough re-write before publication should be considered.

Validity and Major Concerns:

- The authors do not show the inhibitory effect on substrate transport by using a transport assay, which makes the peptide only a potential inhibitor. A transport assay is required to provide the necessary proof for transport inhibition.

- Solubilization in a mixture of DDM/CHS accounts for the well-established stabilizing effect of sterols on ABC-transporters from the plasma-membrane. It is necessary to consistently use the same detergent-environment across all experiments. As both enzyme activity and transport have shown to be responsive to the used detergent, the authors are required to perform their ligand-binding and ATPase activity measurements consistently within the same detergent as for their cryo-EM studies.

Originality & significance:

ABCC5 or MDR5 is linked to multidrug-resistance in cancer and viral infections, effectively reducing drug efficiently through efflux from targeted cells. Biomedical intervention through the development of novel MDR5 inhibitors is of considerable interest, but requires a better understanding of its structure and mechanism of action. The authors provide compelling evidence for a new class of peptide inhibitors against hMDR5 by addressing the intrinsic auto-inhibition-site of the ABC transporter.

Data & Methodology:

The section on MD-simulation and mass-spectrometry are outside the scope of my expertise, thus making me unable to assess fully.

Conclusions:

The presented data largely supports the interpretations provided by the authors. The below listed concerns have to be addressed:

1. L55: MRP5 might work “differently” – This passage needs to be more specific in terms of the ABCC family. Please do compare briefly what is known about the family in terms of a canonical mechanism.
2. L144: Speculation about the relevance of non-conserved motifs and link to a “one-of-a-kind action” needs to be moved to the discussion section. A comparison to the hMRP5 with orthologs would strengthen the termed “uniqueness” of the discussed sequence motifs.
3. L156: To fully show the effect of the hMDR5-deltaR mutant, it is necessary to test its ATPase activity.
4. L169: MD-Simulations to mimic the ER-membrane only represents the synthesis and trafficking path of MRP5 and not its final destination within the plasma-membrane. Mol-percentages of charged PIP-Lipids and overall cholesterol-contents largely differ between ER and Plasma-membrane. It is necessary to compare protein lipid interactions by using different membrane-systems to represent the shift of the lipid code accordingly. This might even show a re-orientation of the R domain.
5. L204: The idea of localization fine-tuning sounds very compelling. Retention of protein-transport based on post-translational modifications has been observed. Since glycosylation takes place in the Golgi, glycosylation-dead mutants would provide additional insight into the mechanism of localization fine-tuning. There might be a link between protein-lipid interaction and protein-glycosylation.

6. L276: Please correct “D35C-E35C” to D35C-E75C.
7. L284: Comparison and a list of the KD values is completely missing. Please add the missing data. Are the commonly used MRP5-inhibitors more specific (lower KD) than the novel peptide-based inhibitors? If there are no known values, the authors should test binding affinities for comparison. Suitable candidates would be cisplatin or doxorubicin.
8. L294: The novel peptide inhibitor needs to be tested in terms of affecting ATPase activity with basal activity and addition of substrates like cisplatin. Alternatively, growth-resistance assays under substrate-addition (like cisplatin) with subsequent addition of the novel peptide should be performed to confirm its inhibitory function. Only after showing an inhibitory effect, the peptide can truly be termed an inhibitor. Consequently, L294-295 cannot be supported by the data.
9. L333: The auto-inhibitory peptide dissociates from the binding pocket. Please search alternative positions by revisiting the density for unassigned regions and comment on potential findings.
10. L342: The passage about conformational changes between the apo and auto-inhibited state requires an additional description on where these residues are located.
11. L359: Major helix-re-arrangements during IF to OF transition like bending of TM5 have been observed for other ABC-transporter like ABCB1. Please add a passage, that includes these findings.
12. L405: Please include an analysis of conformational and compositional heterogeneity by using established tools like OccuPy (<https://doi.org/10.1038/s41467-023-41478-1>) to support this statement.
13. L407: The authors propose a dynamic equilibrium between the auto-inhibited and uninhibited state, which cannot be supported by the resolved wt structure. Further classification needs to be performed to separate the proposed transient states.
14. L411: Poly-specificity of the substrate-binding site might as well explain the binding of different peptides. Many ABC-transporters associated with multi-drug resistance show properties of poly-specific substrate binding sites. Please include this aspect.
15. L439: The authors used hMRP5-delta-1-94 for MST-based ligand-binding studies, mentioned a low affinity for several established ligands but did not show the data. The authors need to include the mentioned data and perform the experiment with the wt protein.
16. L451: The authors mentioned a high specificity of their peptide-inhibitors against hMRP5 without testing their peptide against other members of the ABCC family. Without testing for specificity, the data does not justify the use of inhibitory specificity.
17. L464: De-novo peptide design with high affinity and specificity can benefit from novel bioinformatic tools like RFdiffusion (<https://doi.org/10.1038/s41586-023-06415-8>). Please include this possibility when discussing novel peptide design.
18. L518: The cryo-EM structures were resolved in digitonin, but ATPase activity was measured in DDM/CHS. It is necessary to test enzyme activity in the conditions chosen for cryo-EM.

19. L549: Ligand affinity was determined in DDM/CHS, but your cryo-EM experiments were performed with digitonin-solubilized samples. The authors need to analyze ligand binding under the same conditions as their structural studies.
20. Fig. 2 a: bMRP1 or hMRP1? in the MSA.
21. Fig. 2 b: Units and values of electrostatic potential missing. What is the importance of the highlighted residues, please clarify in the caption.
22. Fig. 2 d: Blotting-results suffer from low contrast. Please adjust for visibility.
23. Fig. 2 e: Add appropriate statistics and uncertainties to the colocalization bar-graphs. Include areas of interest with increased magnification.
24. Fig. 3 a: Please justify the use of different – very similar - contour levels for protein-regions. This seems unnecessary for what you are trying to show. Please add a similar view of the cryo-EM density for wt and delta-1-94 focusing on the binding pocket.
25. Fig. 3 d,f: Annotated residues are almost non-recognizable. Please adjust for clarity.
26. Fig. 3 e: Please add annotations for the electrostatic potential surface and include units & values.
27. Fig. 4 b: Adjust font-size and scale for visibility. List KD for all tested peptides.
28. Fig. 4 d: Please justify the use of different contour levels for protein-regions. This seems unnecessary for what you are trying to show. Please add a measured opening angle of the annotated extracellular gate.
29. Fig. 5 a: Mentioned ligands are poorly visible. Please adjust accordingly.
30. Extended Data Fig. 1 c: bMRP1 or hMRP1? in the MSA.
31. Extended Data Fig. 2-5,7 b: The chosen color-range depicting the resolution-range for the local resolution estimation needs to be adjusted to show local differences.
32. Extended Data Fig. 6: Please adjust to pair-wise structure comparison to make differences visible.
33. Extended Data Fig. 8 e: Building of ATP and Mg²⁺ at about 4 Å resolution is highly debatable. All model-interpretations should reflect the limitations of the derived model from 4 Å reconstructions.
34. Extended Data Fig. 9 c: Since the model was built from a 4 Å reconstruction, the local density should be included to better visualize the limitations of the given model.
35. Supplementary Table 1: Refer to AlphaFold and Phyre as “Initial model used”.

Reviewer #2 (Remarks to the Author):

In the manuscript entitled “Transport mechanism and peptide inhibitor of the first autoinhibited ABC transporter, hMRP5”, Ying Huang, et al., study the mechanism of human multidrug resistance protein 5 (hMRP5). Based on their finding of an autoinhibitory peptide, the authors designed a peptide blocking substrate entry into the transporter. In the study, structural, biochemical, computational, and cell biological technics were combined to provide a mechanistic understanding of hMRP5.

The expertise of this reviewer is in the field of computational modeling, so the review focusses on the computational aspects of the manuscript. Molecular dynamics simulations were applied to study the stability of the autoinhibitory fragment and the flexibility and membrane interactions of the R motif. Overall, the models underlying the atomistic and coarse-grained molecular dynamics simulations seem to be suitable. However, there are several issues described in detail below, which should be addressed in a major revision before the manuscript is suitable for publication in Nature Communications.

- 1) Most importantly, the sampling and analysis of the coarse-grained simulations should be extended to clearly show the preference for negatively charged PA lipids. A single 6 μ s simulation was performed (line 586) at coarse-grained resolution, which corresponds to a comparatively low sampling. Please increase the sampling and analyze the convergence of the discussed protein-lipid interactions.
- 2) In Figure 1b, the lines and symbols are hard to distinguish. Please use different colors to facilitate recognition of the different experimental conditions.
- 3) Line 174 ff: Please quantify the lipid-protein interactions from the coarse-grained simulations. From the snapshot displayed in Figure 2c, a preferred interaction of the R motif with PA lipids is not evident.
- 4) Line 179 ff: Why is solely binding to singly phosphorylated PIs observed but not higher phosphorylated PIs? Is it possible to obtain any structural information from the MD simulations?
- 5) Line 215: Please provide more details about the building of the atomic model of Met1-Thr94.
- 6) Figure 3c, top: Which part of the protein was used for fitting before the calculation of the RMSD of C46-S64 peptide?

7) Line 225: The total simulation time of the atomistic simulations was 1 μ s, so please replace “microsecond time scales” by “the total simulation time of 1 μ s” to avoid confusion.

8) Line 576: Using Martini 3, no EIneDyn model is available for proteins following the standard protocols. There are solely a standard elastic network (which is different from EIneDyn) or a Go-like model available. Please explain more details about the EIneDyn implementation used here or clarify which elastic network was applied.

9) Line 582: Please specify which residues were specifically position restrained. Did this also include residues of the R motif?

10) Supplementary movie 1: In the first half of the simulation, a non-charged residue seems to be strongly interacting with the lipid membrane. Which residue is it? Please discuss its potential role in membrane binding of the R motif.

11) Please provide legends for the supplementary movies.

Reviewer #3 (Remarks to the Author):

This paper presents high-resolution structures for human MRP5 in wild type state, in absence and presence of ATP. Notably, the apo MRP5 structure includes part of the protein in the substrate binding site. Using a series of deletion mutants (hMRP5-delta R, which is missing an insertion within NBD1, and hMRP5-delta1-94, which removes residues N-terminal to the lasso structure), the authors show that residues C46-S64 are bound in the pocket. C46-S64 residues are bound to the pocket through hydrophobic interactions, which were validated by structural analysis of an additional MRP5 mutant in which bulky hydrophobics that bound C46-S64 were altered to Ala, and also by charged and polar interactions. Some of these interactions involved a phosphorylated Ser residue (pS60), which was shown to be phosphorylated by mass spec. The authors solve an additional structure of a complex of hMRP5-delta1-94 with a peptide (D35C-E75C) that was derived from C46-S64, and propose this as a starting point for designing inhibitors of MRP5. The data is clearly presented, in general, and the work showing a peptide bound to the substrate-binding site that could auto-inhibit the transporter expands our understanding of this class of proteins. However, there are some points/thoughts that should be considered by the authors.

Major points

(A) While the appearance of a peptide in the substrate-binding cavity is intriguing and well presented, there are some points that need clarification.

1. Line 141-144: The authors state that a TM5 contains a number of Val residues that are not conserved in other MRPs, and while it is correct that other MRPs do not contain Val at all the positions in which a Val is in TM5 of MRP5, the sequences of MRP5 and other MRPs are quite similar - For example, MRP2 contains 3 of the 6 Val residues. Two of the other Val are replaced by Ile and Thr (Cb-branched amino acids) and a Pro. If the authors wish to discuss this sequence, then perhaps it should be done in the context of the interactions made with the peptide bound in the central cavity.

2. A bound peptide (from the same transporter) bound in the central cavity has been observed in the MRP1 homologue, Ycf1p. Although that paper is only on BioRxiv at the moment, the idea that a disordered element can bind the central cavity may be common to MRPs and thus, the authors may wish to comment on that structure.

3. The authors state that the MRP5I peptide, which binds and blocks the central cavity, inhibits transport. While a transport assay has not been done, on account of the fact the the proteins are in detergent micelles, the authors should consider doing ATPase assays. If the MRP5I peptide is a more potent blocker than the naturally occurring peptide, then presumably decreased ATPase activity would be observed.

4. There are some outstanding questions regarding the model of auto-inhibition. Do the authors have an idea of what would trigger release of the blocking C46-S64 sequence to allow for substrate binding to the central cavity? Is there some specific event or is the interaction transient. Notably, MD simulations show the peptide stably bound in the pocket.

5. The authors assigned C46-S64 as binding the central cavity. A model-in-map fit of this region that includes the side chains should be shown, rather than just the backbone of the model (Figure 1).

6. Finally, the functions of the N-terminal loop (line 51) are not elusive in all ABCC family proteins. Again, this loop has been characterized in CFTR (at least somewhat) as mediating interactions. That study should be referenced.

(B) The following points refer to the R motif that is part of NBD1.

1. The assertion that the R motif is not found in other transporters and/or that its function is not known, is untrue. First, as the authors state, such a motif is seen in MRP9 and MRP11. Although, not as long as the one in MRP5, as the authors state, the MRP9 and MRP11 R motifs are ~30 residues. Further, there is such a motif in CFTR NBD1. Here, it is called the RI; the RI interacts with NBD1, transiently, and phosphorylation of the I alters its NBD1 interactions. A comparison of the MRP5 R motif and the CFTR RI should be made.

2. Additionally, while reading the Results section, I wondered whether the R motif is phosphorylated. The description of the phosphorylation state of the R motif in the Discussion should be moved to the Results section - or at least the phosphorylation state of the R motif should be presented in the Results section.

3. The authors propose that the basic residues in the R motif mediate interactions with negatively charged lipids. However, in light of the fact that at least some of the Ser residues are phosphorylated. How would the observed phosphorylation state affect lipid binding? Further, are the MD simulations done with MRP5 that contains phosphorylated or unphosphorylated R motif. The state should be indicated in the Results section (as well as the Methods section).

Minor points

1. The authors do not call out to every panel in one figure before continuing onto the next. It would ease the readers' job if they could reorder their figures so that all of Figure 1 is called out before Figure 2, etc.

2. The gels shown in Extended Figure 1 are not labeled. The confusion lies with #4 which, if I understand correctly, is the purified hMRP5-deltaR but that has higher MW than the WT or the EQ mutant. Further, #5 seems to be the R motif alone at 15kDa. That seems too high for the R motif and I didn't see anywhere how that region of MRP5 was expressed and purified. Is this a fusion of the R motif?

3. Line 276 - because the Methods are after the results, the authors should state why Cys mutants were made to generate the D35C-E75C peptide. Additionally, because this peptide binds MRP5-delta1-94 with higher affinity, the authors should comment on whether an inter-molecular disulfide bridge is made between the peptide and the transporter.

4. The authors make a comparison of the ATP-bound occluded state of the transporter to ATP-bound structures of SUR1, another ABCC protein. However, why is the comparison not made to MRP1 for which there are apo, ATP-bound, and turnover-conditions structures.

5. The ending sentence of the manuscript is not so much of a conclusion but a statement of future work, making the ending read ore like a grant. The authors may wish to summarize the key findings here.

6. "helixes" should be "helices"

Response to referees' comments

We thank the referees again for their valuable time in reviewing our manuscript and their constructive suggestions. We have carefully taken all of their comments into consideration and prepared a more thorough and clear revision. We hope our revision addresses them all. Please find below a point-by-point response to the referees with our responses in **Blue** and the referees' comments in **Red**.

Reviewer #1 (Remarks to the Author):

Key Results:

The authors used SPA cryo-EM to determine structures of the human ABC transporter ABCC5. In combination with MD-simulations and structure-guided peptide-design, the authors proposed an auto-inhibitory mechanism, which intrinsically restricts transport. Furthermore, they propose, that ABCC5 function might be regulated through interactions between PI(4)P and a sequence-motif of the nucleotide binding domain.

The structures presented in the manuscript are of good to very good quality and so is the research design and the amount of data presented. In particular the authors perform many experiments to validate and identify the interaction of the transporter with the peptide (both as wt and the peptide they developed).

However, there are some major concerns that must be addressed before publication is possible, as outlined in detail below. Most pressing is that the authors do not convincingly show that the peptide really inhibits transport. Secondly, the authors utilize different detergent mixtures for the experiments. It is not unlikely that this biases the data making a comparison difficult.

The manuscript, in its current stage is extremely lengthy and major re-writing is required to make it more concise and to enable the reader to fully grasp its content. The quality of the figures is also not suitable for this journal. The authors need to remove the white space and adhere to their color code throughout. Figure 6 in particular, needs to be redone entirely. Even though the authors use cartoon representation for IF and OF conformations it is not immediately clear which conformation is meant in each specific part. Furthermore, the circular arrangement is not helping to increase clarity.

Also, I suggest that the authors refrain from using terminology like "unique" which appears several times in the manuscript. It may be novel or unprecedented but who knows if it is really unique.

As apparent from the many specific comments below the manuscript requires a thorough re-write before publication should be considered.

Validity and Major Concerns:

- The authors do not show the inhibitory effect on substrate transport by using a transport assay, which makes the peptide only a potential inhibitor. A transport assay is

required to provide the necessary proof for transport inhibition.

• Solubilization in a mixture of DDM/CHS accounts for the well-established stabilizing effect of sterols on ABC-transporters from the plasma-membrane. It is necessary to consistently use the same detergent-environment across all experiments. As both enzyme activity and transport have shown to be responsive to the used detergent, the authors are required to perform their ligand-binding and ATPase activity measurements consistently within the same detergent as for their cryo-EM studies.

Originality & significance:

ABCC5 or MDR5 is linked to multidrug-resistance in cancer and viral infections, effectively reducing drug efficiently through efflux from targeted cells. Biomedical intervention through the development of novel MDR5 inhibitors is of considerable interest but requires a better understanding of its structure and mechanism of action. The authors provide compelling evidence for a new class of peptide inhibitors against hMDR5 by addressing the intrinsic auto-inhibition-site of the ABC transporter.

Data & Methodology:

The section on MD-simulation and mass-spectrometry are outside the scope of my expertise, thus making me unable to assess fully.

Conclusions:

The presented data largely supports the interpretations provided by the authors. The below listed concerns have to be addressed:

Reply:

We thank the referee for recognizing the quality of our structures, research design, and the comprehensive nature of our experimental validation. We understand the concerns raised and have taken substantial steps to address them to enhance our work's clarity, quality, and impact. Below, we address each of the major concerns and comments to clarify how we have amended the manuscript and experimental approach accordingly.

We have undertaken a major revision of the manuscript to improve conciseness and readability. Efforts were made to streamline the content, removing unnecessary details while ensuring that the key findings and conclusions remain clear and accessible to the reader.

The concerns regarding figure quality, particularly for Figure 6, have been addressed. We have revised our figures to eliminate white space, ensure consistent adherence to the color code, and enhance overall clarity. For Figure 6, we revised the cartoon representations and arrangement to make the conformations immediately recognizable and the figure more intuitive.

1.L55: MRP5 might work “differently” – This passage needs to be more specific in

terms of the ABCC family. Please do compare briefly what is known about the family in terms of a canonical mechanism.

Reply:

We thank the referee for his/her constructive suggestions.

We have revised the sentence: “These factors suggest MRP5 might work differently from the well-studied ABCC transporters, including MRP1 (ABCC1) and MRP4 (ABCC4), which begin the transport cycle with an inward-open and competent substrate binding pocket conformation.” (Line 50-53)

2.L144: Speculation about the relevance of non-conserved motifs and link to a “one-of-a-kind action” needs to be moved to the discussion section. A comparison to the hMRP5 with orthologs would strengthen the termed “uniqueness” of the discussed sequence motifs.

Reply:

We thank the referee for his/her comments.

We have moved the part of speculation about the relevance of non-conserved motifs and link to a “one-of-a-kind action” to the discussion section (Line 384-390). Moreover, a comparison among the TM5 kinks of MRP5 among different species was performed (Fig. 1, below), further supporting the uniqueness of TM5, the results of which were also added to the supplementary figure (Supplementary Fig. 1g).

Fig. 1 Sequence comparison of TM5 kink. TM5 sequence alignment among different MRP family (up panel). TM5 sequence alignment among different species of MRP5 (down panel).

3.L156: To fully show the effect of the hMDR5-delta R mutant, it is necessary to test its ATPase activity.

Reply:

We thank the referee for his/her comments.

We have performed the ATPase activity of the hMRP5-delta R mutant, which was lower than the wt-hMRP5, supporting the notion that the R motif may affect the activity of

hMRP5 (Fig. 2, below). Accordingly, we added the ATPase activity result of the hMRP5-delta R mutant in the main figure (Fig. 1c) and main text (Line 130).

Fig. 2 ATPase activity of wt-hMRP5, hMRP5-Δ1-94, hMRP5-ΔR and EQ mutant.

4.L169: MD-Simulations to mimic the ER-membrane only represents the synthesis and trafficking path of MRP5 and not its final destination within the plasma-membrane. Mol-percentages of charged PIP-Lipids and overall cholesterol-contents largely differ between ER and Plasma-membrane. It is necessary to compare protein lipid interactions by using different membrane-systems to represent the shift of the lipid code accordingly. This might even show a re-orientation of the R domain.

Reply:

Thank the referee for his/her constructive feedback. Following the reviewer's recommendation, we have broadened our study to include MD simulations with varied membrane systems, reflecting more accurately the plasma membrane's lipid composition. In the PM-like model, it has an increased cholesterol content of 35% and altered ratios of PI and PS lipids (https://opm.phar.umich.edu/biological_membranes/lipid_composition). Additionally, we substantially prolonged the simulation duration from 6 μ s to 200 μ s for the ER-like membrane model and 100 μ s for the PM-like membrane model to enhance the statistical robustness of our findings.

Our analysis reveals that the replacement of PA by PI lipids and the addition of a higher cholesterol concentration does alter the binding mode of hMRP5's R domain by increasing its binding frequency with the hydrophobic patches and decreasing its binding frequency with the positively charged regions (Fig. 3 below).

Details of the additional simulations and the resulting changes in hMRP5 orientation and dynamics are now included in the revised manuscript (Line 153-168, 598-608, 625-627 and Supplementary Fig. 3).

Fig. 3. (a) Contact frequency of the R-domain interacting with membranes resembling the ER and PM. Arrows in blue point out the regions rich in positively charged residues. The area with positive charge and the pair of hydrophobic areas are marked with lines in blue and green, respectively. (b) Composition of lipids in the simulated ER-like membrane. (c) Composition of lipids in the simulated PM-like membrane.

5.L204: The idea of localization fine-tuning sounds very compelling. Retention of protein-transport based on post-translational modifications has been observed. Since glycosylation takes place in the Golgi, glycosylation-dead mutants would provide additional insight into the mechanism of localization fine-tuning. There might be a link between protein-lipid interaction and protein-glycosylation.

Reply:

We thank the referee for providing an interesting perspective on the mechanism of localization fine-tuning.

As one of the major protein post-translational modifications, glycosylation does affect a wide range of biological processes, including protein intracellular/cell surface localization¹. Accordingly, the R motif was subjected to the online website for glycosylation prediction, the results of which demonstrated that several residues could be modified by O-glycosylation with high potential. However, mass spectrometry analysis verified that the R motif could be phosphorylated but not glycosylated with high potential, ruling out that glycosylation plays a regulatory role in the R motif. Interestingly, MD simulations revealed that hMRP5's R motif could contact with both PM and ER-like membranes; however, the binding frequency with the hydrophobic patches and the positively charged regions varied slightly. The phosphorylation modifications and different binding models between the R motif and membranes fine-tuned the localization of hMRP5. Accordingly, we updated the content in the discussion session: “Moreover, the chemical modification prediction revealed that multiple

residues of the R motif may suffer from phosphorylation and glycosylation (Supplementary Fig. 2g). The subsequent sequence and mass spectrometry analysis verified that the R motif is highly conserved and could be phosphorylated but not glycosylated with high potential, ruling out that glycosylation plays a regulatory role in the R motif. However, it remains unclear how the phosphorylation modifications affect the binding of R motif to lipids.” (Line 406-412)

Interestingly, we have broadened MD simulations of the R motif with varied membrane systems, including plasma membrane (PM) and ER-like membrane. Our analysis revealed that hMRP5's R motif could contact with both PM and ER-like membranes; however, the binding frequency with the hydrophobic patches and the positively charged regions varied slightly (Supplementary Fig. 3). These regulatory insertion and modification sites expand the functional diversity of transporters beyond transmembrane domains, potentially providing a mechanism for localization fine-tuning.

6.L276: Please correct “D35C-E35C” to D35C-E75C.

Reply:

We have corrected “D35C-E35C” to D35C-E75C (Line 248).

7.L284: Comparison and a list of the KD values is completely missing. Please add the missing data. Are the commonly used MRP5-inhibitors more specific (lower KD) than the novel peptide-based inhibitors? If there are no known values, the authors should test binding affinities for comparison. Suitable candidates would be cisplatin or doxorubicin.

Reply:

We thank the referee for his/her comments.

We have re-performed the MST assay using the same buffer as that for preparing cryo-EM samples and updated the figure with determined KD values in main figure (Fig. 4b). Following the reviewer's recommendation, we checked the binding affinity of hMRP5 to trequinsin, a verified hMRP5-inhibitor², and the result showed that the KD value was $42.40 \pm 17.53 \mu\text{M}$ (Fig. 4, below), indicating that trequinsin has a weaker affinity to hMRP5 than M5PI. In addition, we also tested the affinity between MRP5 and doxorubicin, showing the KD value at $53.53 \pm 22.92 \mu\text{M}$ (Fig. 4, below), which is much weaker than that of M5PI. Collectively, M5PI could be more specific than trequinsin and doxorubicin. We updated the revision (Line 255-257, and Fig. 4b).

Fig. 4 Microscale thermophoresis (MST) measurement of the binding affinities of hMRP5- Δ 1-94 to different peptides, trequinsin, as well as doxorubicin.

8.L294: The novel peptide inhibitor needs to be tested in terms of affecting ATPase activity with basal activity and addition of substrates like cisplatin. Alternatively, growth-resistance assays under substrate-addition (like cisplatin) with subsequent addition of the novel peptide should be performed to confirm its inhibitory function. Only after showing an inhibitory effect, the peptide can truly be termed an inhibitor. Consequently, L294-295 cannot be supported by the data.

Reply:

We thank the referee for his/her comments.

To confirm the inhibitory function of the M5PI peptide, we performed the transport assay in *Xenopus laevis* oocytes³, an extensively used system for studying the transport activity of ABC transporters. Briefly, the cRNAs of GFP-tagged hMRP5 and mcherry-tagged M5PI were prepared. *Xenopus laevis* oocytes were injected with 30 nl cRNAs (GFP-tagged hMRP5 and mcherry-tagged M5PI have the equal quantity) or water for three days; then, the oocytes were further incubated with doxorubicin for 180 min. Next, doxorubicin in oocytes was semi-quantified using the mass spectrometry analysis (Fig. 5, below). The result showed that oocytes expressing both hMRP5 and M5PI contained a relatively higher amount of doxorubicin than oocytes expressing only hMRP5, supporting that the M5PI might inhibit hMRP5 transport doxorubicin. We updated the manuscript (Line 257-263, 560-590) and figure (Supplementary Fig. 6a).

Fig. 5 Transport activity of hMRP5. a) GFP fluorescence indicates the expression of genes encoding GFP-tagged hMRP5, and mcherry fluorescence indicates the expression of genes encoding mcherry-tagged M5PI. Water was used as a blank control. The oocytes used for transporter assay analysis were selected based on the fluorescence level, which indicates the protein expression level. Independent experiments have been repeated three times with similar results. Bar=100 μ m. b) The y axis is the doxorubicin concentration in the final 50 μ l extract of *Xenopus* oocytes for mass spectrometry analysis. Bars are mean \pm SD. Points represent biologically independent experiments ($n \geq 3$). A two-tailed t-test was performed.

9.L333: The auto-inhibitory peptide dissociates from the binding pocket. Please search alternative positions by revisiting the density for unassigned regions and comment on potential findings.

Reply:

We thank the referee for his/her comments.

We had a thorough analysis of the ATP-bound hMRP5 EM map but did not find any densities potentially derived from the auto-inhibitory peptide (Fig. 6, below), indicating the auto-inhibitory peptide dissociated from the binding pocket and was highly flexible. We revised the sentence in main text (Line 304-308): “A thorough analysis of the ATP-bound EM map revealed that no extra EM density was observed in the transport pathway and the remaining regions of hMRP5, supporting that the auto-inhibitory C46-S64 peptide must dissociate from the binding pocket and the whole N-terminal region became very flexible.”

Fig. 6 Structures of wt-hMRP5 and ATP-bound hMRP5. (a) cryo-EM map and cartoon representation of wt-hMRP5. C46-S64 in the binding pocket was circled in red. (b) cryo-EM map and different sliced side views of ATP-bound hMRP5.

10.L342: The passage about conformational changes between the apo and auto-inhibited state requires an additional description on where these residues are located.

Reply:

We thank the referee for his/her constructive comments.

More detailed location information of the re-orientated residues was added in the main text, showing as follows: “The side chain orientations of several residues differed, including Tyr244 (TM2), Glu289 (TM3), Phe452 (TM6), Phe988 (TM9), Lys1040 (TM10), and Arg1148 (TM12).” (Line 316)

11.L359: Major helix-re-arrangements during IF to OF transition like bending of TM5 have been observed for other ABC-transporter like ABCB1. Please add a passage, that includes these findings.

Reply:

We thank the referee for his/her suggestions.

We have added a passage to describe the helix-re-arrangements during the IF to OF transition for ABCB1 transporter. “Notably, the unique kink in the middle of TM5 disappeared as TM5 became a straight helix (Fig. 5b), reminiscent of the kinked TM4 and TM10 of ABCB1⁴, which were re-arranged to a straighter conformation during the apo to ATP-bound transition”. (Line 333-336)

12.L405: Please include an analysis of conformational and compositional heterogeneity

by using established tools like OccuPy (<https://doi.org/10.1038/s41467-023-41478-1>) to support this statement.

Reply:

We thank the referee for his/her helpful comments.

Accordingly, we used OccuPy to analyze the conformational and compositional heterogeneity of wt-hMRP5 (Fig. 7, below); results revealed that the auto-inhibitory peptide has a high heterogeneity as compared to the surrounding residues, indicating its relative flexibility. We revised the sentences in the discussion session as follows: “While the C46-S64 peptide seems fixed in the binding pocket through multiple interactions, its density in the cryo-EM map is weaker than surrounding residues mainly because of its high heterogeneity (Supplementary Fig 1e).” (Line 377-379)

Fig. 7 wt-hMRP5 map, annotated by the occupancy, and colored according to the estimated local scale.

13.L407: The authors propose a dynamic equilibrium between the auto-inhibited and uninhibited state, which cannot be supported by the resolved wt structure. Further classification needs to be performed to separate the proposed transient states.

Reply:

We thank the referee for his/her constructive comments.

According to the referee’s suggestion, we performed further classification to capture the putative uninhibited hMRP5 (Fig. 8, below). Unfortunately, all classified 3D maps were in an auto-inhibitory state with approximately the same particle numbers. In combination with the heterogeneity analysis of wt-hMRP5, the auto-inhibitory peptide has a relatively weaker density than surrounding residues, mainly because the auto-inhibitory peptide has a high heterogeneity as compared to the surrounding residues. Therefore, we have removed the sentence “This likely indicates that not all particles used for 3D reconstruction were in the auto-inhibited state. We propose that the auto-inhibited and uninhibited conformations exist in a dynamic equilibrium, but quantifying their ratios is challenging” and updated the corresponding sentence in the discussion session as follows “While the C46-S64 peptide is anchored in the binding pocket by

numerous multiple interactions, its density in the cryo-EM map is weaker than surrounding residues mainly because of its high heterogeneity (Supplementary Fig. 1e).” (Line 377-379)

Fig. 8 Three-dimensional classification of wt-hMRP5.

14.L411: Poly-specificity of the substrate-binding site might as well explain the binding of different peptides. Many ABC-transporters associated with multi-drug resistance show properties of poly-specific substrate binding sites. Please include this aspect.

Reply:

We thank the referee for his/her comments.

We have included this aspect in the discussion session accordingly. “Intriguingly, the Asp5-Ser25 peptide also displayed weak hMRP5 affinity, indicating that multiple segments of the N-terminal region are involved in self-interactions with hMRP5. This observation is somewhat similar to what has been seen with ABCB1 and ABCG2, where both proteins have demonstrated multiple specific binding sites.” (Line 380-384)

15.L439: The authors used hMRP5-delta-1-94 for MST-based ligand-binding studies, mentioned a low affinity for several established ligands but did not show the data. The authors need to include the mentioned data and perform the experiment with the wt protein.

Reply:

We thank the referee for his/her comments.

We have added the MST data to show low binding affinities of hMRP5-delta-1-94 and wt-hMRP5 to several substrates (Fig. 9, Fig. 10, below), including progesterone, testosterone, and cGMP. We updated it in the manuscript (Line 429, Supplementary Fig. 6b and 6c).

Fig. 9 Microscale thermophoresis (MST) measurement of the binding affinities between hMRP5-Δ1-94 and different substrates.

Fig. 10 Microscale thermophoresis (MST) measurement of the binding affinities between wt-hMRP5 and different substrates.

16.L451: The authors mentioned a high specificity of their peptide-inhibitors against hMRP5 without testing their peptide against other members of the ABCC family. Without testing for specificity, the data does not justify the use of inhibitory specificity.

Reply:

We thank the referee for his/her comments.

To justify the use of inhibitory specificity against hMRP5, we tested the binding affinity of M5PI peptide to hMRP4 (Fig. 11, below); results showed that the binding affinity between hMRP4 and M5PI could not be estimated through MST test. As a control, PGE1 demonstrated a strong binding affinity to hMRP4 with the KD value at $3.73 \pm 0.66 \mu\text{M}$. Collectively, hMRP4 can rarely capture M5PI, supporting the inhibitory specificity of M5PI against hMRP5. We updated the content accordingly in the revised manuscript (Line 436-439) and figure (Supplementary Fig. 6d).

Fig. 11 (a) Three repeats of microscale thermophoresis (MST) measurement of the binding affinities between hMRP4 and M5PI. (b) MST measurement of the binding affinities between hMRP4 and PGE1.

17.L464: De-novo peptide design with high affinity and specificity can benefit from novel bioinformatic tools like RFdiffusion (<https://doi.org/10.1038/s41586-023-06415-8>). Please include this possibility when discussing novel peptide design.

Reply:

We value the referee's constructive comments. Certainly, our strategy involves the use of RFdiffusion and other AI tools for the de-novo creation of peptides, complemented by biochemical assays and MD simulations to evaluate the inhibitory effects of these peptides. Following the referee's advice, we've updated the discussion section to better illustrate our methodology.

The revised section now states: "We expect that the application of AI technologies, such as RFdiffusion, for the de-novo design of peptides could significantly enhance their pharmacological effects, with their effectiveness subsequently verified via cellular tests." (Line 452-455)

18.L518: The cryo-EM structures were resolved in digitonin, but ATPase activity was measured in DDM/CHS. It is necessary to test enzyme activity in the conditions chosen for cryo-EM.

Reply:

The ATPase activity assay was re-performed under the same digitonin conditions as that for preparing cryo-EM samples; the results displayed a similar pattern to the ATPase activity measured in DDM/CHS. Also, the corresponding data was updated accordingly in the revised version (Line 513-517, Fig. 1c).

19.L549: Ligand affinity was determined in DDM/CHS, but your cryo-EM experiments were performed with digitonin-solubilized samples. The authors need to analyze ligand

binding under the same conditions as their structural studies.

Reply:

We thank the referee for his/her comments.

Yes, the MST assay was also re-performed under the same digitonin conditions as that for preparing cryo-EM samples, displaying a similar binding affinity of ligand to hMRP5. Also, the corresponding data was updated in the revised version (Line 549 and Fig. 4b).

20.Fig. 2 a: bMRP1 or hMRP1? in the MSA.

Reply:

We thank the referee for his/her comments.

Bovine MRP1(bMRP1) shares 91% identity with human MRP1(hMRP1) in the amino acid sequence⁵; however, only the full-length structure of bMRP1 is currently solved. Therefore, we used bMRP1 in the sequence alignment analysis in the first version. To make it clear, we also added the hMRP1 sequence in the sequence comparison in the revised version; please see Figure 2a.

21.Fig. 2 b: Units and values of electrostatic potential missing. What is the importance of the highlighted residues, please clarify in the caption.

Reply:

Units and values of electrostatic potential were added, and the highlighted residues were clarified in the figure legend in the revised version, please see Figure 2b and Figure 3e.

22.Fig. 2 d: Blotting-results suffer from low contrast. Please adjust for visibility.

Reply:

Fig. 2d was updated accordingly in the revised version and the panel was moved to Fig. 2c.

23.Fig. 2 e: Add appropriate statistics and uncertainties to the colocalization bar-graphs. Include areas of interest with increased magnification.

Reply:

Fig. 2e was updated accordingly in the revised version.

24.Fig. 3 a: Please justify the use of different – very similar - contour levels for protein-

regions. This seems unnecessary for what you are trying to show. Please add a similar view of the cryo-EM density for wt and delta-1-94 focusing on the binding pocket.

Reply:

We thank the referee for his/her comments.

Thank the referee for pointing out this mistake, we indeed used the same contour level to show the whole cryo-EM map; we updated the figure legend of Fig. 1d, 3a, and 4d in the revised version (Line 977, 1011 and 1044). In addition, we present a similar view on the binding pocket of wt (Fig. 1d) and Δ 1-94 (Fig. 3a).

25.Fig. 3 d,f: Annotated residues are almost non-recognizable. Please adjust for clarity.

Reply:

Fig. 3 d,f were updated accordingly in the revised version.

26.Fig. 3 e: Please add annotations for the electrostatic potential surface and include units & values.

Reply:

Annotations for the electrostatic potential surface were added, and units & values were added in the revised version.

27.Fig. 4 b: Adjust font-size and scale for visibility. List KD for all tested peptides.

Reply:

Font-size and scale were adjusted, and KD values were listed accordingly in Fig. 4b in the revised version.

28.Fig. 4 d: Please justify the use of different contour levels for protein-regions. This seems unnecessary for what you are trying to show. Please add a measured opening angle of the annotated extracellular gate.

Reply:

Thank the referee for pointing out this mistake; we indeed used the same contour level to show the whole cryo-EM; we updated the figure legend of Fig. 4d in the revised version (Line 1044). Moreover, an angle was added in Fig. 4d in the revised version to describe the open-facing gate accordingly.

29.Fig. 5 a: Mentioned ligands are poorly visible. Please adjust accordingly.

Reply:

Ligands are adjusted accordingly in the revised manuscript.

30. Extended Data Fig. 1 c: bMRP1 or hMRP1? in the MSA.

Reply:

Thank the referee for his/her comments.

Please see the reply in question 20.

31. Extended Data Fig. 2-5,7 b: The chosen color-range depicting the resolution-range for the local resolution estimation needs to be adjusted to show local differences.

Reply:

Thank the referee for his/her positive comments.

We have updated the local resolution maps in the revised version.

32. Extended Data Fig. 6: Please adjust to pair-wise structure comparison to make differences visible.

Reply:

We have supplemented pair-wise structure comparison to make differences visible in the revised version (Supplementary Fig. 5e).

33. Extended Data Fig. 8 e: Building of ATP and Mg²⁺ at about 4 Å resolution is highly debatable. All model-interpretations should reflect the limitations of the derived model from 4 Å reconstructions.

Reply:

Thank the referee for his/her comments.

Since hMRP5 binds ATP and Mg²⁺ through the conserved motifs observed in the ABC transporters, we can build the trustable atomic model of ATP and Mg²⁺ within EM maps even at about 4 Å based on the conserved motifs in published ATP-bound ABC transporter structures^{6,7}. However, the local EM densities of ATP and Mg²⁺ have a relatively poor quality, and the biological significance deduced from the features of ATP and Mg²⁺ of the ATP-bound hMRP5 structures was rarely depicted; thus, we removed Supplementary Fig. 8e and Supplementary Data Fig. 9c, the ATP and Mg²⁺ related interpretations in the revised version.

34. Extended Data Fig. 9 c: Since the model was built from a 4 Å reconstruction, the local density should be included to better visualize the limitations of the given model.

Reply:

Thank the referee for his/her suggestion.

Please see the reply in question 33.

35. Supplementary Table 1: Refer to Alphafold and Phyre as “Initial model used”.

Reply:

Thank the referee for his/her comment.

The parameters used for EM data collecting and processing were summarized in Supplementary Table 1, of which the initial model for 3D reconstruction was generated by the command in cryoSPARC; therefore, the N/A used to describe the initial model is suitable. Instead, we used the structures predicted by Alphafold and Phyre for the atomic model building of hMRP5. To make it clear, we refreshed the Model building passage in the revised manuscript (Line 744-748). Accordingly, we referred to Alphafold and Phyre to build the atomic models in the method and material session.

Reviewer #2 (Remarks to the Author):

In the manuscript entitled “Transport mechanism and peptide inhibitor of the first autoinhibited ABC transporter, hMRP5”, Ying Huang, et al., study the mechanism of human multidrug resistance protein 5 (hMRP5). Based on their finding of an autoinhibitory peptide, the authors designed a peptide blocking substrate entry into the transporter. In the study, structural, biochemical, computational, and cell biological techniques were combined to provide a mechanistic understanding of hMRP5.

The expertise of this reviewer is in the field of computational modeling, so the review focusses on the computational aspects of the manuscript. Molecular dynamics simulations were applied to study the stability of the autoinhibitory fragment and the flexibility and membrane interactions of the R motif. Overall, the models underlying the atomistic and coarse-grained molecular dynamics simulations seem to be suitable. However, there are several issues described in detail below, which should be addressed in a major revision before the manuscript is suitable for publication in Nature Communications.

1) Most importantly, the sampling and analysis of the coarse-grained simulations should be extended to clearly show the preference for negatively charged PA lipids. A

single 6 μ s simulation was performed (line 586) at coarse-grained resolution, which corresponds to a comparatively low sampling. Please increase the sampling and analyze the convergence of the discussed protein-lipid interactions.

Reply:

We appreciate the reviewer's insight regarding the necessity for more comprehensive sampling and analysis in our coarse-grained simulations.

In accordance with your suggestion, we have significantly expanded our simulation timeframe, augmenting it from the initial 6 μ s to an aggregate of 200 μ s. This was accomplished by executing two separate simulations, each with a duration of 100 μ s. This substantial increase in simulation length enhanced the statistical validity of our findings, particularly in the context of the release and rebinding of the R motif. The observed dozens of binding and unbinding events strongly suggest that our system has achieved convergence, as evidenced in the newly included figure (Fig. 12, below). Moreover, our extended data set now clearly demonstrates a marked preference of the hMRP5 R motif for negatively charged PA lipids in the ER membrane (Fig. 3, above; Fig. 13, below).

The manuscript has been updated to incorporate these new findings, including additional figures to present the extended simulations and convergence analysis (Line 153-168, 598-608, 625-627, and Supplementary Fig. 3).

Fig. 12. (a) Trajectories of two coarse-grained molecular dynamics simulations (identified as RUN1 and RUN2) within a ER-like membrane, as a function of the contacts between the R-motif loop and the membrane. (b) Trajectories of two coarse-grained MD simulations within a PM-like membrane.

2) In Figure 1b, the lines and symbols are hard to distinguish. Please use different colors to facilitate recognition of the different experimental conditions.

Reply:

Thank the referee for his/her constructive comments.

We have updated Figure 1b accordingly, using different color lines and symbols and moved to Fig. 1c in the revision.

3) Line 174 ff: Please quantify the lipid-protein interactions from the coarse-grained simulations. From the snapshot displayed in Figure 2c, a preferred interaction of the R motif with PA lipids is not evident.

Reply:

To address the referee's inquiry regarding the quantification of lipid-protein interactions in our coarse-grained simulations, we quantified the interactions between the R motif and lipid phosphate headgroups (Fig. 13, below), utilizing data from our extensive 200 μ s simulations. This analysis distinctly indicates a marked preference of the R motif for PA lipids, evidenced by the highest interaction frequency among the lipid types in an ER-like membrane. Interestingly, we also observed the R motif's ability to interact with other lipid types, such as PC and PE, and notably PS, albeit with lower frequency. This indicates that the R motif's membrane binding affinity is modulated not solely by electrostatic interactions but also by its unique structural features, including a region rich in basic residues and two hydrophobic patches, as depicted in Figures 2a and 2b. These insights have been incorporated into the revised sections of the manuscript, specifically in lines 153-168 and 598-608.

Fig. 13 (a) Composition of lipids in the ER-like membrane used for the coarse-grained molecular dynamics simulations. (b) The observed binding frequency of the R motif with different lipids in the ER membrane.

4) Line 179 ff: Why is solely binding to singly phosphorylated PIs observed but not higher phosphorylated PIs? Is it possible to obtain any structural information from the MD simulations?

Reply:

Thank the referee for raising this important question. Our reasoning behind observing binding predominantly to singly phosphorylated PIs, and not to more highly phosphorylated variants, stems from the unique structural and chemical attributes of these lipid types. While our coarse-grained MD simulations are geared towards exploring the dynamic interactions between the R motif and various lipids (line 598-608, Supplementary Fig. 3), it's crucial to acknowledge that these simulations may lack the detailed chemical resolution needed to differentiate between the various phosphorylation states of PIs. This limitation, despite some reports in the field suggesting the feasibility of such distinctions, means that our current simulations might not fully capture these nuances. We recognize this as an area for future investigation to deepen our understanding of these interactions.

5) Line 215: Please provide more details about the building of the atomic model of Met1-Thr94.

Reply:

We have already added more details of building the Met1-Thr94 atomic model in the methods of manuscript (Line 748-757).

6) Figure 3c, top: Which part of the protein was used for fitting before the calculation of the RMSD of C46-S64 peptide?

Reply:

Apologies for any previous lack of clarity. For further clarification, the RMSD calculation of the C46-S64 peptide was performed using the peptide's own backbone in comparison to the cryo-EM structure. We have made the necessary updates to the figure legends to more accurately represent this detail for clarity (Line 1018-1020).

7) Line 225: The total simulation time of the atomistic simulations was 1 μ s, so please replace “microsecond time scales” by “the total simulation time of 1 μ s” to avoid confusion.

Reply:

We have revised the text to specify that the total duration of the atomistic simulations was 1 μ s, instead of the more general term "microsecond time scales," to prevent any ambiguity (Line 204-205).

8) Line 576: Using Martini 3, no E1NeDyn model is available for proteins following the standard protocols. There are solely a standard elastic network (which is different

from ElnDyn) or a Go-like model available. Please explain more details about the ElnDyn implementation used here or clarify which elastic network was applied.

Reply:

We apologize for the mistake in our manuscript. The simulations we conducted using Martini 3 actually utilized a standard elastic network model, rather than the ElnDyn model as initially stated. We have now corrected this in the manuscript to accurately describe the methodology used in our simulations (Line 610-615).

9) Line 582: Please specify which residues were specifically position restrained. Did this also include residues of the R motif?

Reply:

Thank the referee for his/her query regarding the specific residues that were subjected to position restraints in our study. We appreciate the opportunity to clarify this aspect. The position restraints were not applied to the R motif, primarily because it was not distinctly discernible in the cryo-EM data. The residues that we specifically restrained are from residue 94 to 503, from 572 to 790, and from 840 to 1437. We have made the necessary revisions in the manuscript to ensure this information is presented more clearly and accurately (Line 610-615).

10) Supplementary movie 1: In the first half of the simulation, a non-charged residue seems to be strongly interacting with the lipid membrane. Which residue is it? Please discuss its potential role in membrane binding of the R motif.

Reply:

Thank the referee for the nice question.

In our detailed examination of the protein-lipid interactions within the R motif, as illustrated in Figure 14, below, we identified several non-charged residues instrumental in membrane binding. These are grouped into two hydrophobic regions: the first includes Ala540, Val541, Leu542, and Ala543, while the second is comprised of Leu549, Leu550, and Leu551. Alongside these hydrophobic areas, our findings also highlight occasional yet notable interactions with a series of positively charged residues, namely Lys515, Lys517, Lys518, Lys520, Arg521, Arg524, Lys526, Lys527, Lys529, Arg531, Arg535, with Lys546 and Arg556 positioned between the hydrophobic segments. These charged residues, despite not forming persistent interactions at a consistent site, likely play a pivotal role in facilitating electrostatic engagements with the membrane.

The manuscript has been updated to reflect this comprehensive analysis, specifically

between lines 153-168.

Fig. 14. Contact frequency of the R-domain interacting with the ER-like membrane. Arrows in blue point out the regions rich in positively charged residues. The area with positive charge and the pair of hydrophobic areas are marked with lines in blue and green, respectively.

11) Please provide legends for the supplementary movies.

Reply:

Thank the referee for the suggestions.

The supplementary movie 1 displays a coarse-grained molecular dynamics (MD) simulation of hMRP5 within an ER-like membrane environment. In this visualization, the R motif of hMRP5 is depicted as a green tube for clarity. Additionally, the sidechain beads of Arginine (Arg) and Lysine (Lys) residues are shown as blue sticks. The membrane lipids are illustrated using stick representations, with their phosphate headgroups highlighted as spheres. Furthermore, we have differentiated the headgroups of specific lipid types for better visualization: the headgroups of Phosphatidic Acid (PA) lipids are colored in magenta, while those of Phosphatidylserine (PS) lipids are depicted in red. This color-coding is intended to facilitate a clearer understanding of the interactions between the R motif and various lipid components within the membrane.

The supplementary movie 2 shows an all-atom molecular dynamics simulation of hMRP5 bound to the C46-S64 peptide. The membrane bilayer is depicted through the phosphorus atoms of the lipids. The hMRP5 protein itself is represented as a grey, semi-transparent cartoon. Highlighted within this structure, the R motif is colored green for emphasis. The C46-S64 peptide is illustrated using spheres, which are color-graded from blue at the N-terminal to red at the C-terminal, delineating its orientation. Surrounding residues of hMRP5 are shown in line representation.

Legends have been added to the supplementary movies in our revised submission (Line 1077-1091).

Reviewer #3 (Remarks to the Author):

This paper presents high-resolution structures for human MRP5 in wild type state, in absence and presence of ATP. Notably, the apo MRP5 structure includes part of the protein in the substrate binding site. Using a series of deletion mutants (hMRP5-delta R, which is missing an insertion within NBD1, and hMRP5-delta1-94, which removes residues N-terminal to the lasso structure), the authors show that residues C46-S64 are bound in the pocket. C46-S64 residues are bound to the pocket through hydrophobic interactions, which were validated by structural analysis of an additional MRP5 mutant in which bulky hydrophobics that bound C46-S64 were altered to Ala, and also by charged and polar interactions. Some of these interactions involved a phosphorylated Ser residue (pS60), which was shown to be phosphorylated by mass spec. The authors solve an additional structure of a complex of hMRP5-delta1-94 with a peptide (D35C-E75C) that was derived from C46-S64 and propose this as a starting point for designing inhibitors of MRP5. The data is clearly presented, in general, and the work showing a peptide bound to the substrate-binding site that could auto-inhibit the transporter expands our understanding of this class of proteins. However, there are some points/thoughts that should be considered by the authors.

Major points

(A) While the appearance of a peptide in the substrate-binding cavity is intriguing and well presented, there are some points that need clarification.

1. Line 141-144: The authors state that a TM5 contains a number of Val residues that are not conserved in other MRPs, and while it is correct that other MRPs do not contain Val at all the positions in which a Val is in TM5 of MRP5, the sequences of MRP5 and other MRPs are quite similar - For example, MRP2 contains 3 of the 6 Val residues. Two of the other Val are replaced by Ile and Thr (Cb-branched amino acids) and a Pro. If the authors wish to discuss this sequence, then perhaps it should be done in the context of the interactions made with the peptide bound in the central cavity.

Reply:

Thank the referee for his/her comments.

According to suggestions from referees one and three, we have re-organized the TM5 kink-related information as follows: “Additionally, a notable kink in the middle of TM5 contributes to the formation of the central cavity. This specific TM5 kink is not found in other MRP family members but in their orthologs (Supplementary Fig. 1f and 1g), suggesting its potential role in the translocation of hMRP5 substrates. Collectively, the

unique features of the intrinsic peptide block and the TM5 kink provide key structural insights into the distinct mechanism of hMRP5.’, and moved the content to the discussion session (Line 384-390).

2. A bound peptide (from the same transporter) bound in the central cavity has been observed in the MRP1 homologue, Ycf1p. Although that paper is only on BioRXiv at the moment, the idea that a disordered element can bind the central cavity may be common to MRPs and thus, the authors may wish to comment on that structure.

Reply:

Thank the referee for his/her constructive comments.

We compared the dephosphorylated Ycf1p⁸ on BioRXiv to wt-hMRP5 and gave the comments in the revised main text: “This indicates the central cavity density arose from the Met1-Thr94 flexible region, revealing an auto-blocking mechanism where hMRP5 is inhibited by its own N-terminal fragment. Of note, this is quite different from the Ycf1p (MRP1 ortholog in yeast) structure auto-blocked by the regulatory domain (R-domain) (Supplementary Fig. 4e) positioned between NBD1 and TMD2.” (Line 188-193).

Fig. 15 The atomic model comparison between wt-hMRP5 and dephosphorylated Ycf1p⁸.

3. The authors state that the MRP5I peptide, which binds and blocks the central cavity, inhibits transport. While a transport assay has not been done, on account of the fact the the proteins are in detergent micelles, the authors should consider doing ATPase assays. If the MRP5I peptide is a more potent blocker than the naturally occurring peptide, then presumably decreased ATPase activity would be observed.

Reply:

We thank the referee for his/her comments.

This comment is similar to the comment eight raised by referee one, please see the replies in the corresponding position.

4. There are some outstanding questions regarding the model of auto-inhibition. Do the authors have an idea of what would trigger release of the blocking C46-S64 sequence to allow for substrate binding to the central cavity? Is there some specific event or is the interaction transient. Notably, MD simulations show the peptide stably bound in the pocket.

Reply:

Thank the referee for your insightful question regarding the auto-inhibition model. Our current understanding, partially based on the MD simulations, suggests that the C46-S64 peptide remains stably bound within the pocket, indicating a strong auto-inhibitory interaction. This stable binding implies that the release mechanism might require a specific event or signal rather than being a transient or spontaneous occurrence. While our simulations indicate a stable binding of the C46-S64 peptide, the actual mechanism of its release in a living cell is likely intricate, potentially involving various biochemical or biophysical stimuli that remain to be identified. To further explore this, we plan to conduct additional experimental studies. Specifically, we are in discussions with Ling Wang, a faculty member at our institution with extensive expertise in single-molecule fluorescence spectroscopy, to apply this technique for real-time tracking of the release mechanism of the inhibitory peptide. This approach, inspired by Ling Wang et al.'s use of single-molecule fluorescence spectroscopy to observe conformational changes in bovine MRP1 (bMRP1), promises to shed more light on the intricate dynamics of this auto-inhibition process.

Accordingly, we discussed the released mechanism in the revised manuscript:

“The mechanism by which the inhibitory peptide is released from hMRP5 remains to be elucidated through further investigation. Here, several potential mechanisms for the release of C46-S64 can be proposed. One possibility is that post-translational modifications (PTMs), such as dephosphorylation, could induce conformational changes in the protein structure, weakening the interaction between the C46-S64 sequence and its binding pocket. Alternatively, the binding of allosteric effectors or partner proteins could also serve as a trigger. These molecules could either directly compete for the binding site or induce structural alterations that destabilize the C46-S64 peptide's interaction, thereby facilitating its release.” (Line 391-400)

5. The authors assigned C46-S64 as binding the central cavity. A model-in-map fit of this region that includes the side chains should be shown, rather than just the backbone of the model (Figure 1).

Reply:

Thank the referee for his/her comments.

We have updated the figure to show the side chains; according to the referee's suggestion, we moved the panel to Figure 3b in the revised version.

6. Finally, the functions of the N-terminal loop (line 51) are not elusive in all ABCC family proteins. Again, this loop has been characterized in CFTR (at least somewhat) as mediating interactions. That study should be referenced.

Reply:

Thank the referee for his/her comments.

We have no sense to show that the functions of the N-terminal loop are not elusive in all ABCC family proteins. To make it clear, we have updated the sentence as follows: "However, the biological functions of the N-terminal loop and R motif of hMRP5 remain elusive" (Line 47-49). Moreover, Jue Chen's group has reported structures of ABCC1(MRP1)⁵ and ABCC7 (CFTR)⁹, whose N-terminus starts with the TMD0 (transmembrane domain 0) and lasso domain, respectively. Very recently, our group and other groups reported the structures of ABCC4(MRP4)⁷, whose N-terminus starts with the lasso domain. All these reported structures have no flexible N-terminal loops, which is different from our hMRP5.

Fig 16. a), b), c), and d). Topology diagram of MRP1⁶, CFTR⁹ hMRP4⁷ and hMRP5.

The N terminal was circled in red. [REDACTED]

(B) The following points refer to the R motif that is part of NBD1.

1. The assertion that the R motif is not found in other transporters and/or that its function is not known, is untrue. First, as the authors state, such a motif is seen in MRP9

and MRP11. Although, not as long as the one in MRP5, as the authors state, the MRP9 and MRP11 R motifs are ~30 residues. Further, there is such a motif in CFTR NBD1. Here, it is called the RI; the RI interacts with NBD1, transiently, and phosphorylation of the I alters its NBD1 interactions. A comparison of the MRP5 R motif and the CFTR RI should be made.

Reply:

Thank the referee for his/her comments.

As stated in the discussion session, hMRP8 and hMRP9 each contains a motif in the counterpart region to hMRP5 ; however, both of those motifs are composed of fewer than 50 amino acids, which are shorter than the hMRP5 R motif and lack hydrophobic patches. To make it clear, we also removed the “unique” term about the R motif in the revised manuscript. Of note, the regulatory (R) region/domain in ABCC7/CFTR is located between NBD1 and TMD2, which is quite different from hMRP5’s R motif embedded in NBD1 (Fig. 17, below).

Fig 17. a), b). Topology diagram of MRP5 and CFTR⁹. The R motif of hMRP5 and R domain of CFTR were circled in red. [REDACTED]

2. Additionally, while reading the Results section, I wondered whether the R motif is phosphorylated. The description of the phosphorylation state of the R motif in the Discussion should be moved to the Results section - or at least the phosphorylation state of the R motif should be presented in the Results section.

Reply:

Thank for the comments from referee.

We supplemented the phosphorylation state of the R motif in the results section. (Line

140-144).

3. The authors propose that the basic residues in the R motif mediate interactions with negatively charged lipids. However, in light of the fact that the at least some of the Ser residues are phosphorylated. How would the observed phosphorylation state affect lipid binding? Further, are the MD simulations done with MRP5 that contains phosphorylated or unphosphorylated R motif. The state should be indicate in the Results section (as well as the Methods section).

Reply:

Thank the referee for highlighting the impact of phosphorylation on the interactions between the R motif and negatively charged lipids. Phosphorylation introduces a negative charge to a residue, which could potentially modify its interaction dynamics with negatively charged lipid headgroups. In particular, the addition of negatively charged phosphorylated residues to the R motif may decrease its affinity for these lipid headgroups, leading to changes in binding patterns or preferences. Such alterations could significantly influence the R motif's overall interaction profile with the membrane, including its orientation and the stability of its interactions with specific lipid species.

However, it is important to note that our simulations were conducted using the unphosphorylated form of the R motif, primarily due to the limitations in available force field parameters for phosphorylated residues in the coarse-grained Martini3 model at the time of our study. Consequently, our current simulation data does not directly reflect the potential impact of phosphorylation on the R motif's lipid binding properties. Future studies, incorporating phosphorylation force field parameters in Martini3 model would be required to comprehensively understand the influence of phosphorylation on the R motif's interaction with membrane lipids.

We have updated the manuscript to clarify this aspect more clearly (Line 156-158, 411-412).

Minor points

1. The authors do not call out to every panel in one figure before continuing onto the next. It would ease the readers job if they could reorder their figures so that all of Figure 1 is called out before Figure 2, etc.

Reply:

Thank the referee for his/her suggestion.

We adjusted the order of the diagrams in the revision.

2. The gels shown in Extended Figure 1 are not labeled. The confusion lies with #4 which, if I understand correctly, is the purified hMRP5-deltaR but that has higher MW than the WT or the EQ mutant. Further, #5 seems to be the R motif alone at 15kDa. That seems to high for the R motif and I didn't see anywhere how that region of MRP5 was expressed and purified. Is this a fusion of the R motif?

Reply:

Thank the referee for his/her comments.

We misused the gels of GFP-tagged hMRP5-deltaR, which was used in the localization assay. We updated the gels in the revised version (Fig. 1b and Supplementary Fig. 2f). Besides, the R motif is tagged with C-terminal 2×strep, and the protein size is 12.1 kDa. However, the R motif might suffer from post-translational modifications, which may lead to the protein size being larger. We mentioned the purification process in the methods (Line 505-507).

3. Line 276 - because the Methods are after the results, the authors should state why Cys mutants were made to generate the D35C-E75C peptide. Additionally, because this peptide binds MRP5-delta1-94 with higher affinity, the authors should comment on whether an inter-molecular disulfide bridge is made between the peptide and the transporter.

Reply:

Thank the referee for his/her comments.

The purpose of making Cysteine mutant was to generate a putative cyclized peptide, which would be more stable. MST assay supported that the D35C-E75C peptide bound hMRP5 with higher affinity than the D35-E75 peptide. However, we did not verify the cyclization occurred. It is very interesting to speculate an inter-molecular disulfide bridge was formed between the D35C-E75C peptide and hMRP5. However, there was no density corresponding to an inter-molecular disulfide in our high-resolution structure of the D35C-E75C peptide-bound hMRP5, ruling out the possibility of forming an inter-molecular disulfide. To make it clearer, we also updated the discussion session accordingly (Line 443-446).

4. The authors make a comparison of the ATP-bound occlude state of the transporter to ATP-bound structures of SUR1, another ABCC protein. However, why is the comparison not made to MRP1 for which there are apo, ATP-bound, and turnover-conditions structures.

Reply:

Thank the referee for his/her comments.

We have supplemented the comparison of the ATP-bound hMRP5 to bMRP1 in the revised manuscript (Supplementary Fig. 8e).

5. The ending sentence of the manuscript is not so much of a conclusion but a statement of future work, making the ending read ore like a grant. The authors may wish to summarize the key findings here.

Reply:

Thank the referee for his/her comments.

We have updated the ending sentence in the revised version: “In summary, the auto-blocked structural basis and the newly found biological function of the R motif highly advanced our understanding of the transport mechanism of ABC transporters; moreover, our data provided us a clue for developing peptide drugs specifically targeting hMRP5, shedding light on the treatment of hMRP5 mediated drug resistance.” (Line 455-459)

6. "helixes" should be "helices"

Reply:

Thank the referee for his/her reminder.

We have updated it accordingly in the revised version.

Reference:

1. Pirillo, A., Svecla, M., Catapano, A. L., Holleboom, A. G. & Norata, G. D. Impact of protein glycosylation on lipoprotein metabolism and atherosclerosis. *Cardiovasc. Res.* **117**, 1033–1045 (2021).
2. Jedlitschky, G., Burchell, B. & Keppler, D. The Multidrug resistance protein 5 functions as an ATP-dependent export pump for cyclic nucleotides. *J. Biol. Chem.* **275**, 30069–30074 (2000).
3. Huang, X. *et al.* Cryo-EM structure and molecular mechanism of abscisic acid transporter ABCG25. *Nat. Plants* **9**, 1709–1719 (2023).
4. Nosol, K. *et al.* Cryo-EM structures reveal distinct mechanisms of inhibition of the human multidrug transporter ABCB1. *Proc. Nat. Acad. Sci. U. S. A.* **117**, 26245–26253 (2020).
5. Johnson, Z. L. & Chen, J. Structural basis of substrate recognition by the multidrug resistance protein MRP1. *Cell* **168**, 1075-1085.e9 (2017).
6. Johnson, Z. L. & Chen, J. ATP binding enables substrate release from multidrug resistance protein 1. *Cell* **172**, 81-89.e10 (2018).
7. Huang, Y. *et al.* Structural basis for substrate and inhibitor recognition of human multidrug transporter MRP4. *Commun. Biol.* **6**, 549 (2023).
8. Nitesh, A., Khandelwal, K. & Tomasiak, T. M. Structural basis for autoinhibition by the

dephosphorylated regulatory domain of Ycf1. bioRxiv
doi:<https://doi.org/10.1101/2023.06.22.546176> (2023).

9. Zhang, Z. & Chen, J. Atomic structure of the cystic fibrosis transmembrane conductance regulator. *Cell* **167**, 1586-1597.e9 (2016).

REVIEWER COMMENTS

Reviewer #1 (Remarks to the Author):

The authors have addressed all major points of the reviewers. I have no further concerns.

Reviewer #2 (Remarks to the Author):

I thank the authors for carefully addressing my comments focusing on the computational aspects of the manuscript. The extension of the simulation time to 200 μ s in ER-like inner membrane and addition of 100 μ s simulation in PM-like inner membrane substantially increase the statistics of the lipid binding to the R motif and strengthen the computational analysis presented in the manuscript.

Before publication, the authors should clarify the remaining points concerning Supplementary Figure 3 and the analysis of the lipid binding frequency:

1) In Supplementary Figure 3, the results of the coarse-grained MD simulations are shown. Subfigure b shows the lipid composition of the inner ER-like membrane, while subfigure f shows the binding frequency of the R motif according to the figure caption. However, it seems that both figures show the same lipid composition, such that there would not be any lipid preference of the R motif in the simulations. Please clarify.

2) Subfigure c shows the selectivity for lipids according to the caption, but the y axis label is "Binding Population". Moreover, the y axis does not start at 0 in contrast to the lipid composition of the respective membranes (b and c), which is misleading. Also, calculating the proportion of binding population to their abundance in the membrane, as described in the caption, would yield numbers around 1 according to subfigures f and b. Please clarify.

3) Please include the analysis of lipid binding frequency and lipid preference for the inner PM-like membrane as well.

4) For the ER-like inner membrane model, the cholesterol concentration in both leaflets is asymmetric (outer leaflet 30%; inner leaflet 15%). The tail composition in both leaflets is identical with 100% PO.

Cholesterol might equilibrate to a certain degree between the leaflets due to its rapid flip-flop in coarse-grained simulations. Please ensure that this does not perturb the membrane structure significantly.

Reviewer #3 (Remarks to the Author):

The authors have undertaken a significant amount of work to address the comments of this and other reviewers, including binding studies with peptides and known MRP5 inhibitors, transport assays, and ATPase assays. There are some key points, however, that should be addressed:

1. The authors have not correctly, to my mind, compared the R motif in MRP5 to know analogous region in other ABC proteins of the same subfamily. While they acknowledge that such a motif, albeit shorter, is present in MRP8 and MRP9, they do not mention that such a motif also exists in CFTR that can be phosphorylated like that in MRP5. The analogous region is called the RI (regulatory insert) in CFTR and is distinct from the R domain that connects NBD1 to TMD2. The CFTR RI was first identified in Xray crystal structures of CFTR NBD1, and then subsequently studied using other structural methods and also in functional CFTR experiments. Thus, the added sentences in the paper regarding CFTR need to be altered.

2. Additionally, the N-terminal loop also exists in CFTR and has been shown to be involved in protein interactions, albeit with soluble regions of the protein. The authors may wish to mention this given that the N-terminal loop mediates protein interactions within MRP5.

3. The authors also make a comparison between the auto-inhibited MRP5 structure and the structure of Ycf1p in the dephosphorylated state in which the R domain (that connects NBD1 to TMD2) is found in the substrate cavity in that protein. The sentence on lines 190-193 is somewhat confusing and needs to be modified. When looking at Supplementary Figure 4e, it seems to me that the MRP5 N-terminal loop and the Ycf1p R domain occupy similar positions in the respective transporters - in the central cavity formed by the TMDs - even though they are from different parts of each transporter. The end of the sentence "... blocked by the regulatory domain (R-domain) (Supplementary Fig. 4e) positioned between the NBD1 and TMD2" makes it seem like that is where the R-domain is found in the structure but perhaps the authors were trying to explain where the R-domain is in the sequence.

Minor points:

1. The authors first tested whether density in the central cavity came from the R motif (line 127: "To test this hypothesis ..."). However, no reasoning behind this hypothesis was given, until the following paragraph (starting at line 135). Perhaps the authors may wish to rearrange these two paragraphs for a more natural flow.

2. MD simulations were done in two different lipid environments - PM and ER to look at interactions of the R motif. The authors should comment on the difference in lipid compositions for these two membranes so that the reader can fully evaluate why the R motif binds ER membranes more than PM. The authors allude to the composition differences later, but it would be easier for the reader if a summary of these differences was provided first.

3. GA should be referred to as Golgi or Golgi apparatus to match the figures.

4. The authors may want to consider calling out the relevant figures for the sentences on line 237.

5. Please define PGE1.

Response to referees' comments

We thank the referees again for their valuable time in reviewing our manuscript and their constructive suggestions. We have carefully taken all of their comments into consideration and prepared a more thorough and clear revision. We hope our revision addresses them all. Please find below a point-by-point response to the referees with our responses in **Blue** and the referees' comments in **Red**.

Reviewer #1 (Remarks to the Author):

The authors have addressed all major points of the reviewers. I have no further concerns.

Reviewer #2 (Remarks to the Author):

I thank the authors for carefully addressing my comments focusing on the computational aspects of the manuscript. The extension of the simulation time to 200 μ s in ER-like inner membrane and addition of 100 μ s simulation in PM-like inner membrane substantially increase the statistics of the lipid binding to the R motif and strengthen the computational analysis presented in the manuscript.

Reply:

Thank you for your positive comments and great suggestions for improving our manuscript.

Before publication, the authors should clarify the remaining points concerning Supplementary Figure 3 and the analysis of the lipid binding frequency:

Reply:

We have addressed the issues you raised concerning Supplementary Figure 3 and the analysis of the lipid binding frequency as described below.

1) In Supplementary Figure 3, the results of the coarse-grained MD simulations are shown. Subfigure b shows the lipid composition of the inner ER-like membrane, while subfigure f shows the binding frequency of the R motif according to the figure caption. However, it seems that both figures show the same lipid composition, such that there would not be any lipid preference of the R motif in the simulations. Please clarify.

Reply:

You correctly pointed out our mistake in Supplementary Figure 3f. We apologize for the confusion. In the revised manuscript, the updated Supplementary Figure 3h and 3i show the binding probability of the R motif in both ER and PM membrane, demonstrating its preferential interaction with negatively charged lipids PS, PA and PI than PC and PE, although not specifically to PA. We have further elaborated on this differential lipid interaction in the discussion section (Line 419-426).

2) Subfigure c shows the selectivity for lipids according to the caption, but the y axis

label is “Binding Population”. Moreover, the y axis does not start at 0 in contrast to the lipid composition of the respective membranes (b and c), which is misleading. Also, calculating the proportion of binding population to their abundance in the membrane, as described in the caption, would yield numbers around 1 according to subfigures f and b. Please clarify.

Reply:

Thank you for your careful review. There were indeed a few errors in the original Figure 3. We would assume that you meant Subfigure g rather than c. It was indeed a mistake. We have corrected the label and put the correct figure in the updated manuscript.

We agree that the original y-axis did not start at zero, which could be misconstrued. We have addressed this issue in the revised Figure 3g by setting the y-axis to zero, allowing for a clearer visualization of the relative binding population.

Regarding the lipid preference which was calculated by the ratio of binding population to abundance, we used a distance cutoff to count the lipid interactions with the R motif. The R motif can potentially interact with multiple lipid molecules within that cutoff distance, so the ratio of lipid binding population to their abundance could exceed 1.

We apologize for any confusion caused by the original figure and caption.

3) Please include the analysis of lipid binding frequency and lipid preference for the inner PM-like membrane as well.

Reply:

We have now included the analysis of lipid binding frequency and lipid preference for inner PM-like membrane (now Supplementary Figure 3 g and i). Moreover, we added a paragraph discussing the lipid preference in the discussion section (Line 419-426).

4) For the ER-like inner membrane model, the cholesterol concentration in both leaflets is asymmetric (outer leaflet 30%; inner leaflet 15%). The tail composition in both leaflets is identical with 100% PO. Cholesterol might equilibrate to a certain degree between the leaflets due to its rapid flip-flop in coarse-grained simulations. Please ensure that this does not perturb the membrane structure significantly.

Reply:

We appreciate your vigilance in ensuring the validity of our model. We acknowledge that cholesterol can rapidly translocate between leaflets in coarse-grained simulations. To address this concern, we visually inspected the simulation trajectories generated from the trajectory data (Supplementary Movie 1) which showed that they did not cause significant perturbations to the overall membrane structure within the simulated timescale. We revised the main text accordingly (Line 168-174) and updated Supplementary Movie 1.

Reviewer #3 (Remarks to the Author):

The authors have undertaken a significant amount of work to address the comments of this and other reviewers, including binding studies with peptides and known MRP5 inhibitors, transport assays, and ATPase assays. There are some key points, however, that should be addressed:

1. The authors have not correctly, to my mind, compared the R motif in MRP5 to know analogous region in other ABC proteins of the same subfamily. While they acknowledge that such a motif, albeit shorter, is present in MRP8 and MRP9, they do not mention that such a motif also exists in CFTR that can be phosphorylated like that in MRP5. The analogous region is called the RI (regulatory insert) in CFTR and is distinct from the R domain that connects NBD1 to TMD2. The CFTR RI was first identified in Xray crystal structures of CFTR NBD1, and then subsequently studied using other structural methods and also in functional CFTR experiments. Thus, the added sentences in the paper regarding CFTR need to be altered.

Reply:

Thank you for your constructive suggestions. We have updated the figure presenting the sequence alignment among MRPs and human CFTR (Fig. 2a), and the information on CFTR RI was added in the appropriate position: “Moreover, hMRP5 has a regulatory motif (R motif) in its first nucleotide-binding domain (NBD) (Fig. 1a), and the counterpart region within CFTR NBD1 also contains a similar but shorter amino acid sequence called regulatory insertion (R insertion), whose phosphorylation could serve as a regulator of the conformational switch. However, the biological functions of hMRP5 R motif remain elusive.” (Line 55-60).

2. Additionally, the N-terminal loop also exists in CFTR and has been shown to be involved in protein interactions, albeit with soluble regions of the protein. The authors may wish to mention this given that the N-terminal loop mediates protein interactions within MRP5.

Reply:

We supplemented the information on the N-terminal region in the main text: “Accumulating evidence supports that the N-terminal region (including TMD0 or lasso domain) is involved in the protein interactions and membrane traffic machinery in the ABCC family, indicating that the N-terminal loop is essential for the function of hMRP5.” (Line 51-55).

3. The authors also make a comparison between the auto-inhibited MRP5 structure and the structure of Ycf1p in the dephosphorylated state in which the R domain (that connects NBD1 to TMD2) is found in the substrate cavity in that protein. The sentence

on lines 190-193 is somewhat confusing and needs to be modified. When looking at Supplementary Figure 4e, it seems to me that the MRP5 N-terminal loop and the Ycf1p R domain occupy similar positions in the respective transporters - in the central cavity formed by the TMDs - even though they are from different parts of each transporter. The end of the sentence "... blocked by the regulatory domain (R-domain) (Supplementary Fig. 4e) positioned between the NBD1 and TMD2" makes it seem like that is where the R-domain is found in the structure but perhaps the authors were trying to explain where the R-domain is in the sequence.

Reply:

We apologize for the confusion. Accordingly, we have modified the sentence on lines 190-193 in main text (now on Line 208-211). Moreover, to make it clear, we updated the supplementary Fig. 4e in the revision.

Minor points:

1. The authors first tested whether density in the central cavity came from the R motif (line 127: "To test this hypothesis ..."). However, no reasoning behind this hypothesis was given, until the following paragraph (starting at line 135). Perhaps the authors may wish to rearrange these two paragraphs for a more natural flow.

Reply:

We have already rearranged the paragraph in the main text (starting from line 138 to line 151).

2. MD simulations were done in two different lipid environments - PM and ER to look at interactions of the R motif. The authors should comment on the difference in lipid compositions for these two membranes so that the reader can fully evaluate why the R motif binds ER membranes more than PM. The authors allude to the composition differences later, but it would be easier for the reader if a summary of these differences was provided first.

Reply:

Thank you for the insightful comment. We agree that explicitly outlining the differences in lipid composition between the simulated ER and PM bilayers is crucial for a complete understanding of the R motif's binding behavior. We have incorporated a revised section at the beginning of the results section summarizing the distinct compositions of these membranes and highlighting the presence of more negatively charged lipids (PA) in the ER (Line 168-174 in the revision). This should provide readers with the necessary context to fully appreciate the observed differences in R motif binding.

3. GA should be referred to as Golgi or Golgi apparatus to match the figures.

Reply:

We updated the figure (Figure 2e) and figure legend (main text, Line 1034) to ensure a consistent representation of the Golgi apparatus.

4. The authors may want to consider calling out the relevant figures for the sentences on line 237.

Reply:

We have supplemented the relevant figure (now on line 256).

5. Please define PGE1.

Reply:

We have defined PGE1 in the revision (now on line 463).

REVIEWERS' COMMENTS

Reviewer #2 (Remarks to the Author):

I thank the authors for addressing all my points raised and have no further comments.

Reviewer #3 (Remarks to the Author):

The authors have addressed the additional comments of this reviewer.

Response to referees' comments

We thank the referees again for their valuable time in reviewing our manuscript and their constructive suggestions. We have carefully taken all of their comments into consideration and prepared a more thorough and clear revision. We hope our revision addresses them all. Please find below a point-by-point response to the referees with our responses in **Blue** and the referees' comments in **Red**.

Reviewer #1 (Remarks to the Author):

The authors have addressed all major points of the reviewers. I have no further concerns.

Reviewer #2 (Remarks to the Author):

I thank the authors for addressing all my points raised and have no further comments.

Reviewer #3 (Remarks to the Author):

The authors have addressed the additional comments of this reviewer.

Reply:

Thanks all reviewers for their constructive comments and suggestions on our manuscript.